# FRAP: Faithful and Realistic Text-to-Image Generation with Adaptive Prompt Weighting

**Liyao Jiang**[1,2]**, Negar Hassanpour**[2]**, Mohammad Salameh**[2]**,**
**Mohan Sai Singamsetti**[2]**, Fengyu Sun**[3]**, Wei Lu**[3]**, Di Niu**[1]

[1]*Department of Electrical and Computer Engineering, University of Alberta*

[2]*Huawei Technologies Canada*

[3]*Huawei Kirin Solution, China*

*{liyao1, dniu}@ualberta.ca*

*{negar.hassanpour2, mohammad.salameh, mohan.sai.singamsetti}@huawei.com*

*{sunfengyu, robin.luwei}@hisilicon.com*

**Reviewed on OpenReview:** *https://openreview.net/forum?id=MKCwO34oIq*

## Abstract

Text-to-image (T2I) diffusion models have demonstrated impressive capabilities in generating high-quality images given a text prompt. However, ensuring the prompt-image alignment remains a considerable challenge, i.e., generating images that faithfully align with the prompt's semantics. Recent works attempt to improve the faithfulness by optimizing the noisy image latent code, which potentially could cause the latent code to go out-of-distribution and thus produce unrealistic images. In this paper, we propose FRAP, a simple, yet effective approach based on adaptively adjusting the per-token prompt weights to improve prompt-image alignment and authenticity of the generated images. We design an online algorithm to adaptively update each token's weight coefficient, which is achieved by minimizing a unified objective function that encourages object presence and the binding of object-modifier pairs. Through extensive evaluations, we show FRAP generates images with higher or comparable prompt-image alignment to prompts from complex datasets, while having a lower average latency compared to recent latent code optimization methods, e.g., 4 seconds faster than recent method D&B on the COCO-Subject dataset. Furthermore, through visual comparisons and evaluation of the CLIP-IQA-Real metric, we show that FRAP not only improves prompt-image alignment but also generates more authentic images with realistic appearances. We also explore combining FRAP with prompt rewriting LLM to recover their degraded prompt-image alignment, where we observe improvements in both prompt-image alignment and image quality. We release the code at the following link: `https://github.com/LiyaoJiang1998/FRAP/`.

## 1 Introduction

Recent text-to-image (T2I) diffusion models (Rombach et al., 2022; Chang et al., 2023; Kang et al., 2023) have demonstrated impressive capabilities in synthesizing photo-realistic images given a text prompt. Faithfulness and authenticity are among the most important aspects of assessing the quality of AI-generated content, where faithfulness is evaluated by prompt-image alignment metrics (Radford et al., 2021; Li et al., 2022; Chefer et al., 2023) and authenticity reflects how realistic the image appears and is usually evaluated by image quality assessment metrics (Wang et al., 2023; Wu et al., 2023). While recent T2I diffusion models are capable of generating realistic images with high aesthetic quality, it has been shown (Chefer et al., 2023; Feng et al., 2022; Wang et al., 2022) that the AI-generated images may not always faithfully align with the semantics of the given text prompt. One typical issue that compromises the model's ability to align with the given prompt is catastrophic neglect (Chefer et al., 2023), where one or more objects are missing in the generated image. Another common alignment issue is incorrect modifier binding (Li et al., 2023), where the

modifiers, e.g., color, texture, etc., are either bound to the wrong object mentioned in the prompt or ignored. The generation can also result in an unrealistic mix or hybrid of different objects.

To improve prompt-image alignment, Attend & Excite (A&E) (Chefer et al., 2023) introduces the Generative Semantic Nursing (GSN) method, which iteratively adjusts the latent code throughout the reverse generative process (RGP) by optimizing a loss function defined on cross-attention (CA) maps to improve attention to under-attended objects. Similarly, Divide & Bind (D&B) (Li et al., 2023) uses GSN to improve the binding between objects and modifiers, by designing an optimization objective that aligns the attention of objects and their related modifiers. Although GSN can alleviate the semantic alignment issues, yet the iterative updates to latent code could potentially drive the latent code out-of-distribution (OOD), resulting in less realistic images being generated. In addition, several recent works investigate prompt optimization (Hao et al., 2023; Valerio et al., 2023; Mañas et al., 2024), which aims to improve image aesthetics while trying to maintain the prompt-image alignment, by rewriting the user-provided text prompt with large language model (LLM). Despite improvements in image aesthetic quality, Mañas et al. (2024) show that rewriting prompts primarily for aesthetics enhancement could harm the semantic alignment between the image and prompt. Therefore, it remains a significant challenge to generate images to ensure prompt-image alignment while achieving a realistic appearance and high image authenticity.

In this paper, we introduce Faithful and Realistic Text-to-Image Generation with Adaptive Prompt Weighting (FRAP), which adaptively adjusts the per-token prompt weighting throughout the RGP of a diffusion model to improve the prompt-image alignment while still generating high-quality images with realistic appearance. We utilize the spaCy (Honnibal & Montani, 2017) language parser to identify object tokens and object-modifier relationships from the prompt. Unlike the conventional manual prompt weighting techniques (Hugging Face, 2023a; Stewart, 2023), we design an online optimization algorithm which is performed during inference and updates each token's weight coefficient in the prompt to adaptively emphasize or de-emphasize certain tokens at different time-steps, by minimizing a unified objective function defined based on the cross-attention maps of the identified object and modifier tokens to strengthen the presence of the prompted object and encourage modifier binding to objects. Since we optimize each prompt token's weight coefficient, our approach avoids the above-mentioned out-of-distribution issue associated with directly modifying the latent code. Furthermore, we also introduce a novel usage of FRAP by combining it with LLM-based prompt optimizer to achieve outstanding enhancement to both prompt alignment and generation quality.

We extensively evaluate the faithfulness, overall image quality, and image authenticity of FRAP-generated images via prompt-image alignment metrics, image quality assessment metrics, and an image authenticity metric. We make the following observations:

- We observe that FRAP achieves higher or comparable faithfulness than recent methods on all prompt-image alignment metrics on the Color-Obj-Scene, COCO-Subject, and COCO-Attribute datasets (Li et al., 2023) which contain complex and challenging prompts, while remaining on par with these methods on simple datasets with prompts created from templates. In the meantime, the images generated by FRAP demonstrate significantly higher authenticity with a more realistic appearance in terms of the CLIP-IQA-Real measure (Wang et al., 2023).

- While improving prompt-image alignment and generating more realistic and authentic images, our method requires lower average inference latency (e.g., 4 seconds faster than D&B on the COCO-Subject dataset) and lower number of UNet calls by not making repeated calls of UNet at each time-step during inference, in contrast to the recent GSN methods (Chefer et al., 2023; Li et al., 2023) which calls the UNet multiple times in a refinement loop at each time-step, especially on datasets with complex prompts that require many repeated calls.

- We observe that although the LLM prompt rewriting method Promptist (Hao et al., 2023) improves the image aesthetic quality, it could harm the semantic alignment between the image and prompt. Moreover, our adaptive prompt weighting approach can be easily integrated with recent prompt rewrite methods (Hao et al., 2023; Valerio et al., 2023; Mañas et al., 2024). We explore applying FRAP to the rewritten prompts of Promptist to recover their degraded prompt-image alignment, where we observe improvements in both the prompt-image alignment and overall image quality.

## 2 Preliminaries

### 2.1 Stable Diffusion (SD)

We implement our approach based on the open-source T2I latent diffusion model SD (Rombach et al., 2022). Unlike traditional diffusion models (such as Sohl-Dickstein et al. (2015); Ho et al. (2020); Song et al. (2020)) that work in the pixel space, SD operates in an autoencoder's latent space, which significantly reduces the amount of compute resources required. SD is composed of several components, including a Variational Auto-Encoder (VAE) (Kingma & Welling, 2013; Rezende et al., 2014)[1], a UNet (Ronneberger et al., 2015) that operates in the latent space and serves as the backbone for image generation, and a Text Encoder that is used to obtain embeddings from the text prompt.

#### 2.1.1 UNet

For training the UNet, the forward diffusion process (Ho et al., 2020) is used to collect a training dataset. Noise $\epsilon$ is gradually added to the original latent $z_0$ over $T+1$ time-steps[2], and the info corresponding to a subset of time-steps $t$ is recorded in a form of $\{t, z_t, \epsilon_t\}$. Rombach et al. (2022) trained a UNet (Ronneberger et al., 2015) denoiser, parameterized by $\theta$, to predict the added noise $\epsilon_t$ conditioned on the latent code $z_t$, the time-step $t$, and the text embedding $c$ from the text encoder:

$$\mathcal{L}_{\text{train}} = \mathbb{E}_{z \sim \mathcal{E}(x), \, y, \, \epsilon \sim \mathcal{N}(\mathbf{0},\mathbf{I})} \left[ ||\epsilon - \epsilon_\theta(z_t, t, c)||_2^2 \right]. \tag{1}$$

To generate novel images conditioned on text prompt, the already-trained UNet is called to sequentially denoise a latent code sampled from the unit Gaussian distribution $z_T \sim \mathcal{N}(\mathbf{0}, \mathbf{I})$ and conditioned on the text embedding $c$, by traversing back through $T+1$ time-steps following the RGP. The denoising process results in producing a clean final latent code $z_0$, which is passed to the VAE's decoder to obtain the respective generated image $\mathcal{I} = \mathcal{D}(z_0)$.

#### 2.1.2 Cross-Attention (CA)

The CA unit is responsible for injecting the prompt information into the model. A CA map at time-step $t$ is denoted by $A_t$, having a $L \times L \times N$ tensor size, where $L$ is the spatial dimension of the intermediate feature map and $N$ is the number of prompt tokens. The CA layer weights project the text embedding $c$ into keys $K$ and values $V$, while queries $Q$ are mapped from the intermediate feature map of the UNet. Subsequently, the CA map $A_t$ is calculated as $A_t = Softmax(\frac{QK^T}{\sqrt{d}})$.[3] We represent the CA map for token $n$ by $A_t^n = A_t[:,:,n]$. We utilize the same Gaussian filter $G$ from Chefer et al. (2023) to smooth the CA map (i.e., $G(A_t^n)$) for each token. Smoothing the CA maps ensures that the influence of neighboring patches is also reflected in the probability of each patch.

#### 2.1.3 Text Encoder

Conditioned on the text prompt $y$, the CLIP (Radford et al., 2021) text encoder produces an $N \times H$ text embedding tensor $c^y$, where $c^y = \text{CLIP}(y) = [c^{y,1}, c^{y,2}, ..., c^{y,N}]$, $N$ denotes the number of tokens, and $H$ is the dimension of the text embedding. To satisfy the requirements for the classifier-free guidance (CFG) (Ho & Salimans, 2022) approach, an unconditional text embedding is also generated from the null text $\varnothing$, where $c^\varnothing = \text{CLIP}(\varnothing) = [c^{\varnothing,1}, c^{\varnothing,2}, ..., c^{\varnothing,N}]$. The conditional token embedding for token $n$ is denoted as $c^{y,n} \in \mathbb{R}^H$ and the unconditional token embedding for token $n$ is denoted as $c^{\varnothing,n} \in \mathbb{R}^H$.

### 2.2 Prompt Weighting

Prompt weighting (Hugging Face, 2023a; Stewart, 2023) is a popular technique among SD art creators for obtaining more control over the image generation process. Users can manually specify per-token weight

---

[1]The VAE encoder $\mathcal{E}$ maps an input image $x$ into its spatial latent representation $z = \mathcal{E}(x)$, while the decoder $\mathcal{D}$ attempts to reconstruct the same image from $z$ such that $\mathcal{D}(\mathcal{E}(x)) \approx x$.

[2]Often, $T$ is chosen to be a large number (e.g., 1000) such that $z_T$ is close to the unit Gaussian noise.

[3]As such, $A_t[i, j, :]$ defines a probability distribution over the tokens.

coefficients $\phi = [\phi^1, \phi^2, ..., \phi^N]$ to emphasize or de-emphasize the semantics behind each token. Prompt weighting obtains a weighted text embedding $\tilde{c^y}$ by interpolating between the conditional text embedding $c^y$ and the unconditional text embedding $c^\varnothing$ with the weight coefficients $\phi$ as follows:

$$\tilde{c^y}(\phi, c^y, c^\varnothing) = \phi \odot c^y + (1 - \phi) \odot c^\varnothing, \tag{2}$$

where $\odot$ denotes, e.g., multiplication of each scalar $\phi^i$ by token $i$'s embedding vector $c^{y,i}$.

The SD model adopts the CFG (Ho & Salimans, 2022) approach to control the strength of conditional information on the generated image. This strength is controlled with a guidance scale $\beta$. With the prompt weighted text embedding $\tilde{c^y}$, CFG updates the UNet's predicted noise as follows:

$$\tilde{\epsilon}_\theta(z_t, t, c) = \beta\epsilon_\theta(z_t, t, \tilde{c^y}) + (1 - \beta)\epsilon_\theta(z_t, t, c^\varnothing). \tag{3}$$

The manual prompt weighting technique (Hugging Face, 2023a; Stewart, 2023) uses fixed weight coefficients throughout the entire RGP. Moreover, for every single prompt, the users must manually determine the weight coefficients through expert knowledge.[4]

## 3 Related works

T2I generation has been most successful with the recent advancement in diffusion models (Sohl-Dickstein et al., 2015; Ho et al., 2020). These include large-scale models such as Stable Diffusion (SD) (Rombach et al., 2022), Imagen (Saharia et al., 2022), MidJourney (Midjourney.com, 2023), and Dall·E 2 (Ramesh et al., 2022). Although these diffusion models have adopted methods such as classifier guidance (CG) (Dhariwal & Nichol, 2021) and CFG (Ho & Salimans, 2022), it remains a challenge for them to enforce proper alignment with some prompt scenarios (Chefer et al., 2023; Feng et al., 2022; Wang et al., 2022; Marcus et al., 2022). Several engineering techniques have been developed to partially address this shortcoming (Liu & Chilton, 2022; Marcus et al., 2022; Witteveen & Andrews, 2022).

A recent line of work attempts to automate this task by developing methods for training-free improvement of the semantics compliance of SD, which is an open-sourced text-to-image generation diffusion model. In the remainder of this section, we describe the most prominent related works in the literature and touch on their potential limitations.

StructureDiffusion (Feng et al., 2022) used a language model to obtain hierarchical structures from the prompt in the form of noun phrases. They then used the average of the respective CA maps (Hertz et al., 2022) as the output of the CA unit. This method, however, is limited in terms of its ability to generate images that diverge significantly from those produced by SD (Rombach et al., 2022).

Attend & Excite (A&E) (Chefer et al., 2023) introduced the novel concept of GSN, which refers to manipulating the latent codes to guide the RGP towards improving the image alignment with the prompt. They proposed a loss function based on the CA maps to update the latent codes at each time-step. However, their loss function does not explicitly consider the modifier binding issue. Moreover, it only considers a point estimate (i.e., the highest pixel value among all CA maps of noun objects), making it susceptible to having robustness issues.

Divide & Bind (D&B) (Li et al., 2023) adopted the GSN idea (Chefer et al., 2023), but proposed a novel composite loss function that addresses both catastrophic neglect with a Total Variation (TV) loss, as well as the modifier binding issue with a Jensen Shannon Divergence (JSD) loss.

SynGen (Rassin et al., 2023) employed a language model (spaCy's transformer-based dependency parser (Honnibal & Montani, 2017)) to syntactically analyze the prompt. This was to first identify entities and their modifiers, and then use CA maps to create an objective that encourages the modifier binding only. As a result, if the latent code misses the generation of an object, their binding-only loss cannot compensate for this error.

---

[4]The manual prompt weighting library (Stewart, 2023) suggests a bounded range of $\phi \in [0.5, 1.5]$, since too large or too small of a weighting coefficient has been empirically shown to create low-quality results.

## 4    Method

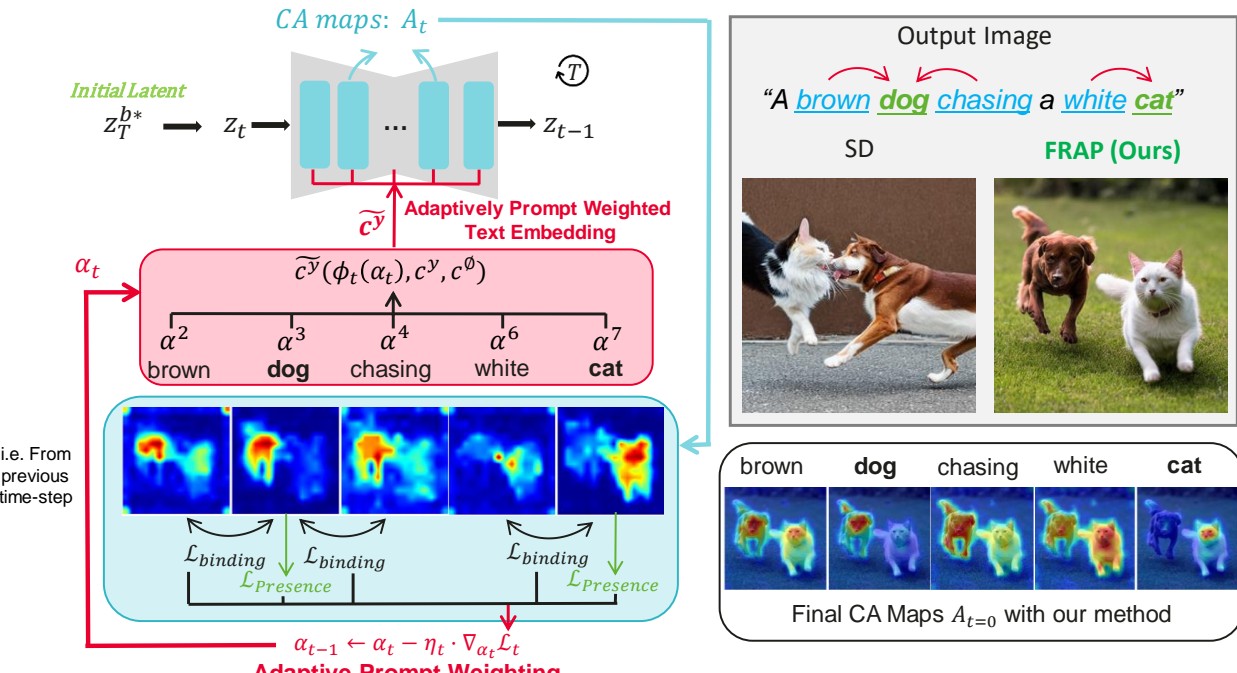

Figure 1: **Method overview.** We perform adaptive updates to the per-token prompt weight coefficients to improve faithfulness. The updates are guided by our unified loss function, defined on the CA maps, which strengthens object presence and encourages object-modifier binding. FRAP is capable of correctly generating every object along with accurate object-modifier bindings.

Prior works (Chefer et al., 2023; Li et al., 2023) improve the semantic guidance in SD by optimizing the latent code during inference. These methods demonstrate excellent capability in addressing semantic compliance issues. However, the latent code could go out-of-distribution (OOD) after several iterative updates, which could lead to generation of unrealistic images.

To address this issue, we propose a simple, yet effective approach based on the prompt weighting technique that improves faithfulness while maintaining the realistic appearance of generated images. Unlike the manual approach, FRAP adaptively updates the weight coefficients of different text tokens during the RGP to emphasize or de-emphasize certain tokens in order to facilitate improving faithfulness. The weight updates are guided by our unified objective function defined on the CA maps, strengthening the object presence and encouraging object-modifier binding.

The following subsections describe our proposed unified objective function and adaptive prompt weighting method. The overview of FRAP is illustrated in Fig. 1. In App. A, we present the overall algorithm and implementation details of our proposed method. We include our code in the supplementary material and will open-source our codebase for further research after review.

### 4.1    Unified Objective

We propose an objective function $\mathcal{L}$, formulated based on the CA maps, to enhance the alignment between the generated image and the prompt. This unified objective function is used as the loss for on-the-fly optimization of the adaptive prompt weight parameters. Our proposed unified objective function $\mathcal{L}_t$ at time-step $t$ consists of two terms: (i) the object presence loss $\mathcal{L}_{\text{presence},t}$ and (ii) the object-modifier binding loss $\mathcal{L}_{\text{binding},t}$. The overall objective function is defined as:

$$\mathcal{L}_t = \mathcal{L}_{\text{presence,t}} - \lambda \mathcal{L}_{\text{binding},t}, \tag{4}$$

where $\lambda$ is a hyperparameter.

To automate the manual identification of object tokens and modifier tokens from the prompt, we follow SynGen (Rassin et al., 2023) and use the spaCy (Honnibal & Montani, 2017) language parser to extract the set $S$ of all object tokens $s \in S$ and set $R$ of all modifier tokens $r \in R$ in the prompt. For each object token $s$, we use $R_s$ to represent its related modifier tokens $r \in R_s$. We use $(s, r)$ to represent an object-modifier pair (e.g. "brown dog"), where $P$ is the pairing operation and $P(s, R_s)$ represents the set of all object-modifier pairs between object token $s$ and its related modifier tokens $r \in R_s$. We use $P(S, R)$ to represents the set of all object-modifier pairs identified in the prompt. The object presence loss is defined on the set of object tokens $S$ in the text prompt to reinforce the presence of each object $s \in S$ in the generated image to prevent neglect or generating hybrid creatures. The object-modifier binding loss is defined on the set of all object-modifier token pairs $(s, r) \in P(S, R)$ to encourage the correct binding of each object token $s \in S$ and each of its related modifier token $r \in R_s$. More details on object and modifier token identification are explained in App. A.3.

### 4.1.1 Object Presence Loss

Our object presence loss aims to enhance the significance of each object token $s \in S$. We achieve this by maximizing the largest activation value of the Gaussian smoothed CA map $G(A_t^s)$ for each object token. We define the loss for $s$ at time-step $t$ as:

$$\mathcal{L}_{\text{presence},t}^s = 1 - \max\bigl(G(A_t^s)\bigr). \tag{5}$$

As pointed out in the ablation studies done by Chefer et al. (2023), the Gaussian smoothing applied to the CA map is important, since it ensures that maximizing the largest activation value not only encourages a single point of high activation but rather a region of high activation values, which is critical for generating the entire object and avoiding partial generation of objects.

Then, we calculate the total presence loss as the mean of per-token object presence loss for all object tokens $s \in S$:

$$\mathcal{L}_{\text{presence},t} = \frac{1}{|S|} \sum_{s \in S} \mathcal{L}_{\text{presence,t}}^s. \tag{6}$$

Note that the *per-token* presence loss in Eq. (5) is adopted from A&E's (Chefer et al., 2023) design. However, our design of the *total* object presence loss in Eq. (6) is original and different from A&E. We use the mean among all objects as the total object presence loss instead of only using the loss from the most neglected token with the lowest max activation. Our design enhances the presence of *all* objects instead of a single one, which is expected to be essential for complex prompts. Through our empirical evaluation in Table 6 in App. C, our design which enhances the presence of all objects outperforms A&E's design where only the most neglected token is considered.

We also consider an alternative design for the object presence loss, where we attempt to replace the object presence loss with the Total Variation (TV) loss adopted by D&B (Li et al., 2023), which encourages high activation difference across spatial locations. Through our empirical evaluation in Table 7 in App. C, the object presence loss in Eq. (5) performs better than the TV loss.

### 4.1.2 Object-Modifier Binding Loss

The loss function for object-modifier binding aims to ensure that each object is correctly aligned with its corresponding modifier tokens in terms of spatial positioning. To achieve this, the probability densities of each pair $(s, r) \in P(s, R_s)$ should overlap as much as possible. We define the binding loss as follows:

$$\mathcal{L}_{\text{binding},t}^{s,r} = \frac{1}{L^2} \sum_{\text{pixels}} \left( \min\Bigl(\Pr\bigl(G(A_t^s)\bigr), \Pr\bigl(G_a(A_t^r)\bigr)\Bigr) \right), \tag{7}$$

where $L$ is the spatial dimension of the CA map, $G_a(A_t^r)$ denotes the aligned smoothed CA map of the token $r$ according to token $s$ following Li et al. (2023), and $\Pr\bigl(G(A_t^s)\bigr)$ is the normalized discrete probability

distribution of the smoothed CA map of token $s$. This distribution is obtained by applying Softmax over all pixels of the respective CA map. We apply the same process to get $\Pr\big(G_a(A_t^r)\big)$.

For the total object-modifier binding loss, we compute it as the mean of all pairwise object-modifier binding loss $\mathcal{L}_{\text{binding},t}^{s,r}$ for all $(s,r) \in P(S,R)$:

$$\mathcal{L}_{\text{binding},t} = \frac{1}{|P(S,R)|} \sum_{(s,r) \in P(S,R)} \mathcal{L}_{\text{binding,t}}^{s,r}. \tag{8}$$

Finally, we calculate the total overall loss $\mathcal{L}_t$ at time-step $t$ according to Eq. (4).

As shown in Eq. (4), we maximize this minimum overlap between the two discrete probability distributions in a minimax fashion.[5] Our minimax approach aims to ensure that the least probable events have high probabilities in both distributions. This guarantees that every pixel of their respective object or modifier receives a considerable probability, regardless of which CA map is more accurate overall. This is particularly useful in cases where there is a need to depict a small object in the scene. We found this approach to be more beneficial than Minimizing JSD (as in D&B (Li et al., 2023)) or KLD (as in SynGen (Rassin et al., 2023)), likely due to the following two reasons: (i) it *ensures a robust match between the CA maps.* This is particularly useful for representing small objects in a scene. (ii) it is *less susceptible to the influence of outliers* compared to divergence methods such as JSD and KLD. This robustness is beneficial in handling noisy and uncertain CA maps. Through our empirical evaluation in Table 8 in App. C, our object-modifier binding loss in Eq. (7) performs better than either the JSD loss or the KLD loss.

## 4.2 Adaptive Prompt Weighting

Different from the manual prompt weighting (Hugging Face, 2023a; Stewart, 2023), we propose FRAP, an adaptive prompt weighting method, to update the weight coefficients during the RGP automatically. We update the per-token weight coefficients $\phi$ after each time-step $t$ according to our unified loss objective $\mathcal{L}_t$ (see Eq. (4)) defined on the internal feedback from the CA maps. Our approach determines the per-token weights automatically and appropriately for each token to encourage the SD (Rombach et al., 2022) model to generate images that are faithful to the semantics and have a realistic appearance.

In adaptive prompt weighting, we define learnable per-token weight parameters $\alpha = [\alpha^1, \alpha^2, ..., \alpha^N]$ and use the Sigmoid function $\sigma$ to obtain the weight values $\phi$ bounded within $[\phi_{LB}, \phi_{UB}]$ via:

$$\phi = \phi_{LB} + (\phi_{UB} - \phi_{LB})\sigma(\alpha). \tag{9}$$

During the RGP, we perform on-the-fly optimization to the weight parameters $\alpha_t$ according to the gradient $\nabla_{\alpha_t} \mathcal{L}_t$ after each time-step $t$. Then, we use the updated weight parameters $\alpha_{t-1}$ to strengthen the object presence and encourage better object-modifier binding in the next time-step $t-1$:

$$\alpha_{t-1} = \alpha_t - \eta_t \cdot \nabla_{\alpha_t} \mathcal{L}_t, \tag{10}$$

where $\eta_t$ is a scalar step-size for the gradient update. We obtain the bounded next time-step per-token weight coefficients $\phi_{t-1}$ from $\alpha_{t-1}$ using Eq. (9).

At time-step $t = T$, we initialize $\alpha_T = \vec{0}$ so we start from a trivial weighting of $\phi_T = \vec{1}$ (equivalent to the vanilla SD with no prompt weighting). Then, our on-the-fly optimization will gradually adjust the weight coefficients to improve alignment with the prompt. This optimization step takes place during inference with no additional overhead for model training. We only perform one UNet inference call at each time-step, and we use the weight parameter $\alpha_t$ obtained from the optimization in the previous time-step $t+1$ to get the current weight coefficient $\phi_t$. Thus, our adaptive prompt weighting process keeps the same number of UNet calls as the vanilla SD (Rombach et al., 2022). In contrast, other methods such as A&E (Chefer et al., 2023) and D&B (Li et al., 2023) perform several optimization refinement steps at each time-step.

---

[5]Note the negative sign for $\mathcal{L}_{\text{binding},t}$ in Eq. (4), which results in maximizing the minimum overlap.

# 5 Experiments

In this section, we provide quantitative evaluation and visual comparisons of the effectiveness of FRAP in terms of prompt-image alignment and image quality assessment. We also assess the efficiency of FRAP. In addition, we apply FRAP to the recent prompt rewrite method. Moreover, we perform human evaluations in addition to using automated evaluation metrics.

In App. C, we present an ablation study on the design choices of FRAP, including a comparison to alternative loss functions. In App. D, we perform evaluations on additional model, baselines, and even larger datasets. Moreover, we provide additional visual comparisons in Figs. 13 and 14 in the appendix.

**Experimental Settings.** We compare our results to SD (Rombach et al., 2022) and to recent methods A&E (Chefer et al., 2023) and D&B (Li et al., 2023). All baselines and our approach are evaluated based on the SDv1.5 (Rombach et al., 2022; Hugging Face, 2023b) model to align with D&B (Li et al., 2023).[6] We evaluate on three *Simple*, manually crafted prompt datasets from A&E: Animal-Animal (S-AA), Color-Object (S-CO), and Animal-Object (S-AO); and five *Complex* datasets from D&B: Animal-Scene (C-AS), Color-Object-Scene (C-COS), Multi-Object (C-MO), COCO-Attribute (C-CA), and COCO-Subject (C-CS). For comparing the efficiency of different methods, we measure the end-to-end latency and total number of UNet calls. In the experiments, our method utilizes the unified objective function on early time-step cross-attention maps to efficiently select an initial latent code with decent object presence and object-modifier binding. We obtain the initial latent code by performing 15 initial inference steps on a batch of 4 noisy latent code samples and select one with minimal loss. In App. A, we further elaborate on the baselines, implementation details, prompt datasets, evaluation metrics, and other experimental settings.

## 5.1 Prompt-Image Alignment, Overall Image Quality, and Image Authenticity

We use multiple metrics to evaluate the performance of various methods in three different aspects: prompt-image alignment, overall image quality, and image authenticity. For prompt-image alignment, we evaluate metrics including Text-to-Text Similarity (TTS) (Chefer et al., 2023; Li et al., 2023), Full-prompt Similarity (Full-CLIP) (Chefer et al., 2023), and Minimum Object Similarity (MOS) (Chefer et al., 2023). Each metric reflects how well the generated image aligns with the given prompt at a different semantic level. TTS (Chefer et al., 2023; Li et al., 2023) evaluates the sentence-to-sentence level alignment, using the BLIP (Li et al., 2022) captioning model to generate a caption for the synthesized image and compares the cosine similarity between the CLIP (Radford et al., 2021) embeddings of the original prompt and the generated caption of the synthesized image. Full-CLIP (Chefer et al., 2023) reflects the sentence-to-image level alignment by measuring the cosine similarity between CLIP embeddings of the original text prompt and CLIP embeddings of the synthesized image. MOS (Chefer et al., 2023) considers a finer-grained alignment between the sub-sentence (i.e. part of the prompt) and the synthesized image. To evaluate the overall image quality (i.e. image sharpness, brightness), we utilize the Human Preference Score version 2 (HPSv2) (Wu et al., 2023) which is trained on human preference datasets and the CLIP-IQA (Wang et al., 2023) metric which is overall CLIP-IQA score averaged over four criteria including "quality", "noisiness", "natural", and "real". For image authenticity, we adopt the "real" component of the CLIP-IQA metric (i.e. CLIP-IQA-Real) which specifically reflects the level of image authenticity and how realistic the generated images are. See App. A.3 for more detailed description of the evaluation metrics.

We have placed the detailed performance over all datasets and evaluation metrics in the larger Table 5 in App. B. Summarizing the results here, we observe a tight competition among A&E, D&B, and FRAP across all metrics in datasets with *simple* hand-crafted templates (S-AA, S-CO, and S-AO), while all three methods exceed the performance of vanilla SD. This shows that the *simple* datasets may not be sufficient to definitively identify the superior method among the competing options due to the simplicity of the prompts.

We see a clear advantage of FRAP on more challenging *complex* datasets. In Table 1, FRAP achieves either comparable or improved overall image quality in terms of HPSv2 and CLIP-IQA while having a lower latency than A&E and D&B. More importantly, FRAP achieves significant gain in CLIP-IQA-Real score on

---

[6]Note, however, that FRAP is also applicable to SDXL (Podell et al., 2023) and improves its performance, as evident by our additional experiments (see Table 15 in App. D).

the Color-Obj-Scene, COCO-Attribute datasets and consistently achieves the highest performance in image authenticity reflected by the CLIP-IQA-Real metric. Whereas the prior methods could potentially drive the latent code out-of-distribution (OOD) causing the low image authenticity reflected by lower CLIP-IQA-Real scores despite having decent prompt-image alignment and overall image quality (see Fig. 3 where images generated by A&E and D&B have cartoonish appearances in "turtle", "horse", and "clock"). We also updated Table 1 to include statistical significance t-test results. For prompt-image alignment, FRAP significantly improves TTS score and Full-CLIP score on the Color-Obj-Scene and COCO-Attribute datasets which are the more challenging datasets since they include modifier words to the object words, the improvement on these two datasets signifying that FRAP handles object-modifier binding well. Whereas FRAP achieves comparable prompt-image alignment on the simpler COCO-Subject dataset which only contains object words without any modifiers, making it easier for all methods to only deal with object presence without handling any object-modifier binding. We further discuss the observed limitations in App. E, where all evaluated methods (FRAP and all other methods including Standard SD (Rombach et al., 2022), A&E (Chefer et al., 2023), and D&B (Li et al., 2023)) fail to generate the correct number of objects on the Multi-Object dataset (e.g., the count of generated objects does not match the prompt).

Table 1: **Quantitative comparison** of FRAP and other methods on three *complex* datasets Color-Object-Scene, COCO-Attribute, and COCO-Subject. Higher is better (except for latency). Our method is efficient, for instance, FRAP is on average 4 and 5 seconds faster than D&B and A&E respectively, for generating one image on the COCO-Subject dataset. * means that FRAP's improvement over the next best method is statistically significant at $p < 0.05$ with a t-test.

| Dataset | Method | Prompt-Image Alignment | | | Overall Image Quality | | Image Authenticity | Latency (s) ↓ |
|---|---|---|---|---|---|---|---|---|
| | | TTS ↑ | Full-CLIP ↑ | MOS ↑ | HPS v2 ↑ | CLIP-IQA ↑ | CLIP-IQA Real ↑ | |
| Color-Obj-Scene (Li et al., 2023) | SD (2022) | 0.707 | 0.366 | 0.254 | 0.282 | 0.675 | 0.770 | **8.0** |
| | A&E (2023) | 0.708 | 0.376 | 0.267 | 0.286 | 0.675 | 0.770 | 17.9 |
| | D&B (2023) | 0.719 | 0.375 | 0.264 | 0.286 | **0.682** | 0.808 | 16.7 |
| | FRAP (ours) | **0.727*** | **0.381*** | **0.269*** | **0.287** | 0.676 | **0.835*** | 14.3 |
| COCO-Attribute (Li et al., 2023) | SD (2022) | 0.817 | 0.347 | 0.245 | 0.286 | 0.703 | 0.750 | **8.1** |
| | A&E (2023) | 0.820 | 0.353 | **0.254** | 0.286 | 0.703 | 0.750 | 15.3 |
| | D&B (2023) | 0.821 | 0.353 | 0.251 | 0.288 | 0.720 | 0.760 | 16.7 |
| | FRAP (ours) | **0.838*** | **0.358*** | **0.254** | **0.289** | **0.725** | **0.778*** | 13.9 |
| COCO-Subject (Li et al., 2023) | SD (2022) | 0.829 | 0.331 | 0.250 | 0.278 | 0.805 | 0.852 | **7.9** |
| | A&E (2023) | 0.829 | 0.333 | 0.254 | 0.277 | 0.805 | 0.852 | 19.1 |
| | D&B (2023) | 0.835 | 0.333 | 0.253 | **0.279** | **0.814** | 0.862 | 18.1 |
| | FRAP (ours) | **0.837** | **0.334*** | **0.255** | **0.279** | **0.814** | **0.866** | 14.0 |

In addition, we evaluate the FID (Fréchet Inception Distance) metric (Heusel et al., 2017) which is a full-reference IQA metric that requires a ground-truth reference image in addition to the generated image, different from the no-reference IQA metrics (i.e. HPSv2 (Wu et al., 2023) and CLIP-IQA (Wang et al., 2023)) which only requires the generated image. Therefore, we follow the evaluation settings of prior works in text-to-image generation (Rombach et al., 2022; Saharia et al., 2022; Ramesh et al., 2022; Podell et al., 2023) and adopt the validation set of the MS-COCO dataset (Lin et al., 2014) which contains the ground-truth reference image. To select prompt-image pairs that are most relevant to prompt-image alignment evaluation and specifically object presence and object-modifier binding, we only keep the relevant prompts with at least two objects and each has at least one modifier word by filtering the prompts with the spaCy (Honnibal & Montani, 2017) language dependency parser. This filtering process selects 16k most relevant prompts from the original 40k prompts in MS-COCO, and we randomly sample a 5k subset from the 16k most relevant prompts. We refer to this dataset as COCO-5K and will release this dataset to facilitate reproducibility and further research.

In Table 2, we observe significant improvements and best performance for FID (the lower the FID, the better) and all other evaluated metrics for prompt-image alignment, overall image quality, and image authenticity for the COCO-5K dataset. Especially, there is a substantial improvement in CLIP-IQA-Real score over

the three other evaluated methods showing the advantage of achieving high image authenticity even when optimizing for prompt-image alignment when using our proposed FRAP method.

Table 2: **Quantitative comparison** of FRAP and other methods on the COCO-5K subset from the MS-COCO (Lin et al., 2014) dataset. Higher is better (except for FID and latency). Our method is efficient, for instance, FRAP is on average 7 and 10.1 seconds faster than D&B and A&E respectively, for generating one image on the COCO-5K dataset. * means that FRAP's improvement over the next best method is statistically significant at $p < 0.05$ with a t-test (note that the t-test does not apply to FID since it is a distance measure between two distributions with no standard deviation).

| Method | FID ↓ | Prompt-Image Alignment | | | Overall Image Quality | | Image Authenticity | Latency (s) ↓ |
|---|---|---|---|---|---|---|---|---|
| | | TTS ↑ | Full-CLIP ↑ | MOS ↑ | HPS v2 ↑ | CLIP-IQA ↑ | CLIP-IQA Real ↑ | |
| SD (2022) | 39.52 | 0.801 | 0.322 | 0.247 | 0.276 | 0.712 | 0.784 | **4.2** |
| A&E (2023) | 35.82 | 0.799 | 0.324 | 0.250 | 0.275 | 0.705 | 0.782 | 21.3 |
| D&B (2023) | 39.08 | 0.802 | 0.323 | 0.248 | 0.276 | 0.713 | 0.785 | 18.2 |
| FRAP (ours) | **34.22** | **0.811*** | **0.327*** | **0.251*** | **0.279*** | **0.750*** | **0.849*** | 11.2 |

## 5.2 Efficiency Comparisons

Latency values are reported in the last column of Table 1 and also Table 5 in App. B. FRAP enjoys a lower average latency compared to the recent methods, as it does not rely on costly refinement loops. For instance, FRAP is on average 4 seconds faster than D&B and 5 seconds faster than A&E for generating one image on the C-CS dataset. Our reported latency measures the average wall-clock time for generating one image on each dataset in seconds with a V100 GPU. Moreover, our method is also more efficient in terms of the number of UNet calls. FRAP consistently use 66 UNet calls which include 15 calls for initial selection steps and 51 calls for adaptive prompt weighting steps. In contrast, A&E and D&B *at least* perform 76 UNet calls which include 25 time-steps with repeated latent code updates each requiring two UNet calls and another 26 normal time-steps each requiring one UNet call. This number will increase for A&E and D&B with more repeated refinement steps needed on more complex prompts at each time-step.

## 5.3 Integration with Prompt Rewrite

In addition to the training-free methods, we also consider the recent training-based prompt optimization method Promptist (Hao et al., 2023) which collects an initial dataset and finetunes an LLM to rewrite user prompts. Note that it requires expensive training, and it needs to run the LLM alongside the T2I diffusion model. In Table 3, our results show that although Promptist improves the image aesthetics, it significantly degrades the prompt-image alignment, demonstrated by the lower alignment scores than SD using the original user prompt. FRAP's adaptive prompt weighting can easily integrate with prompt rewrite methods and could be applied to the rewritten prompt to recover their degraded prompt-image alignment. By applying FRAP on the rewritten prompt of Promptist, we observed improvements in both the prompt-image alignment and overall image quality over the Promptist method as shown in Table 3. In Fig. 2, we provide visual comparisons of FRAP, the LLM prompt rewrite method Promptist (Hao et al., 2023), and the result of combining FRAP and Promptist.

## 5.4 Visual Comparisions

### 5.4.1 Realistic Appearance and Aesthetics

In Fig. 3, we observe that SD (Rombach et al., 2022) ignores one of the objects (e.g., "crown", "turtle", "dog") in all prompts and often incorrectly binds the color modifiers, yet it maintains a good level of authenticity. Both A&E and D&B generate less realistic images (e.g., "turtle", "horse", and "clock" images exhibit cartoonish appearance), likely due to the aforementioned OOD issue. In contrast, FRAP generates images that are more faithful to the prompt, containing all the mentioned objects and well-separated with

Table 3: **Quantitative comparison** of SD, training-based LLM prompt optimization/rewrite method Promptist (Hao et al., 2023), applying FRAP on Promptist rewritten prompt, and FRAP, on the *complex* Color-Object-Scene dataset.

| Method | Prompt-Image Alignment | | | Overall Image Quality | | Image Authenticity | Latency (s) ↓ |
|---|---|---|---|---|---|---|---|
| | TTS ↑ | Full-CLIP ↑ | MOS ↑ | HPS v2 ↑ | CLIP-IQA ↑ | CLIP-IQA Real ↑ | |
| SD (2022) | 0.707 | 0.366 | 0.254 | 0.282 | 0.675 | 0.770 | 8.0 |
| Promptist (2023) | 0.666 | 0.358 | 0.249 | 0.279 | 0.698 | 0.939 | 9.2 |
| Promptist (2023) + FRAP | 0.694 | 0.377 | 0.265 | 0.283 | **0.702** | **0.940** | 15.5 |
| FRAP | **0.727** | **0.381** | **0.269** | **0.287** | 0.676 | 0.835 | 14.3 |

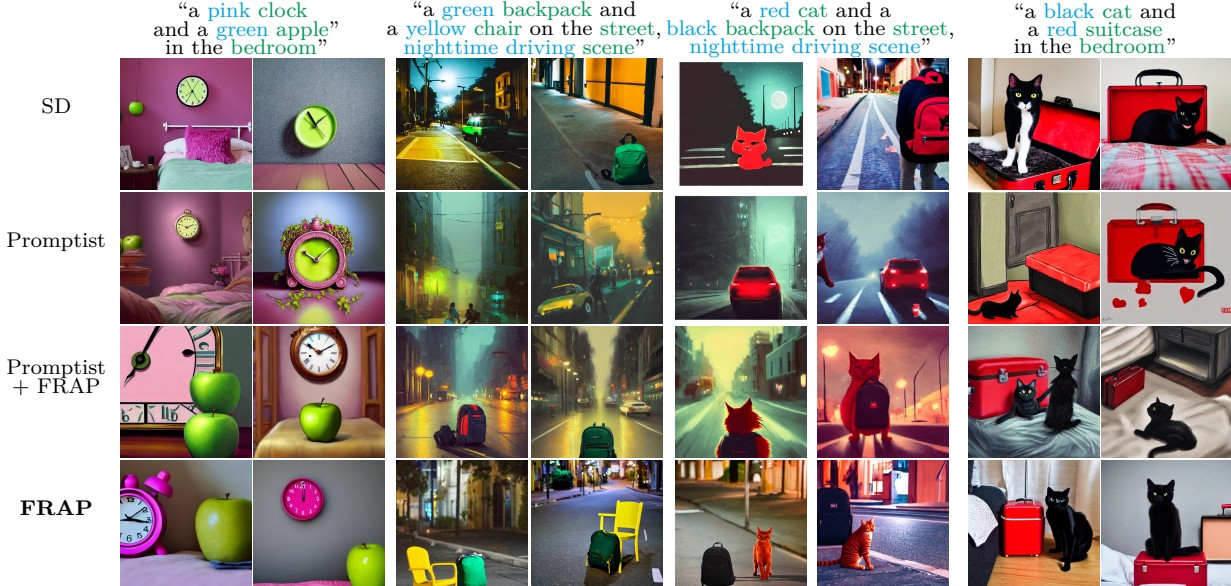

Figure 2: **Qualitative comparison** of SD, training-based LLM prompt rewrite method Promptist (Hao et al., 2023), applying FRAP on Promptist rewritten prompt, and FRAP, on the *complex* Color-Object-Scene dataset. For each prompt, we show images generated by all four methods (using the same set of seeds). The object tokens are highlighted in green and modifier tokens are highlighted in blue.

accurate color bindings (e.g., "clock" and "apple" with correct color bindings instead of a hybrid of two objects as seen for other methods). More importantly, images generated with FRAP maintain a more realistic appearance.

### 5.4.2 Object Presence and Object-Modifier Binding

In Fig. 4, we observe that SD, A&E, and D&B exhibit object neglect issues on the top four prompts from simple datasets, such as missing "frog", "bow", and "dog". Also, they often generate a hybrid of two objects, such as a "bear" with frog-like feet, a bear-shaped "balloon", and a toy-like "elephant". In addition, these images have incorrect color bindings on the prompts. FRAP, however, generates more realistic images and mitigates object neglect, incorrect binding, and hybrid object issues. Additional visual comparisons are provided in Figs. 13 and 14 in the appendix.

### 5.4.3 Complex Prompts

The bottom four sets of images in Fig. 4 pertain to more complex prompts (C-COS, C-CS, C-CA, and C-MO datasets) which include more objects and more complicated scenes with modifiers. For example, for

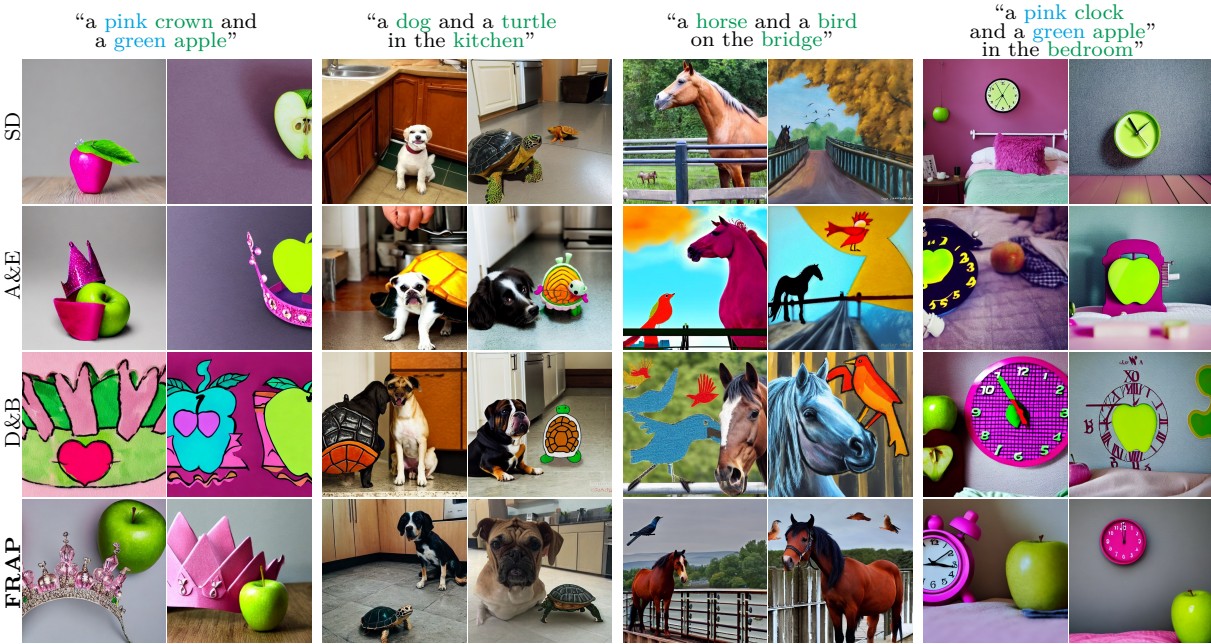

Figure 3: **Qualitative comparison.** Stable Diffusion (Rombach et al., 2022) often exhibits both issues of (i) catastrophic neglect, and (ii) incorrect attribute binding. A&E (Chefer et al., 2023) and D&B (Li et al., 2023) directly modify the latent code to improve faithfulness. However, the latent code could go out-of-distribution after several iterative updates, resulting in lower quality and less realistic images despite improving its semantics. Our FRAP method can generate images that are faithful to the prompt while maintaining a realistic appearance and aesthetics.

"a green backpack and a yellow chair on the street, nighttime driving scene", all three baselines are missing the "yellow chair" or mixing "yellow" with the "road". On the prompt "a black cat sitting on top of a green bench in a field", the bench is either neglected or does not have the correct green color. Conversely, FRAP generates images that remain faithful to the prompt in these more complex scenarios.

## 5.5 Human Evaluations

In addition to quantitative evaluation metrics, we perform human evaluations by randomly selecting 60 prompts from the 4 complex datasets (i.e. C-COS, C-CS, C-CA, and C-AS datasets). For each prompt, we ask 6 participants to choose the best image among the outputs of SD, A&E, D&B, and FRAP. When presenting each prompt, we randomly shuffle the order of the images from different methods to ensure an unbiased evaluation. FRAP is voted as the majority winner in 42.1% of the test-cases and significantly outperformed others (i.e. SD with 10.5%, A&E with 22.8%, and D&B with 24.6%), which also aligns with our quantitative findings using the automated evaluation metrics. We demonstrate the interface of the human evaluation in Fig. 5.

## 6 Conclusions

In this work, we propose FRAP, an adaptive prompt weighting method for improving the prompt-image alignment and authenticity of images generated by diffusion models. We design an online algorithm for adaptively adjusting the per-token prompt weights, which is achieved by minimizing a unified objective function that strengthens object presence and encourages object-modifier binding.

We conduct extensive experiments and demonstrate that FRAP achieves higher or comparable faithfulness than latent code optimization methods such as A&E (Chefer et al., 2023) and D&B (Li et al., 2023) on all prompt-image alignment metrics on the Color-Obj-Scene, COCO-Attribute, and COCO-Subject datasets

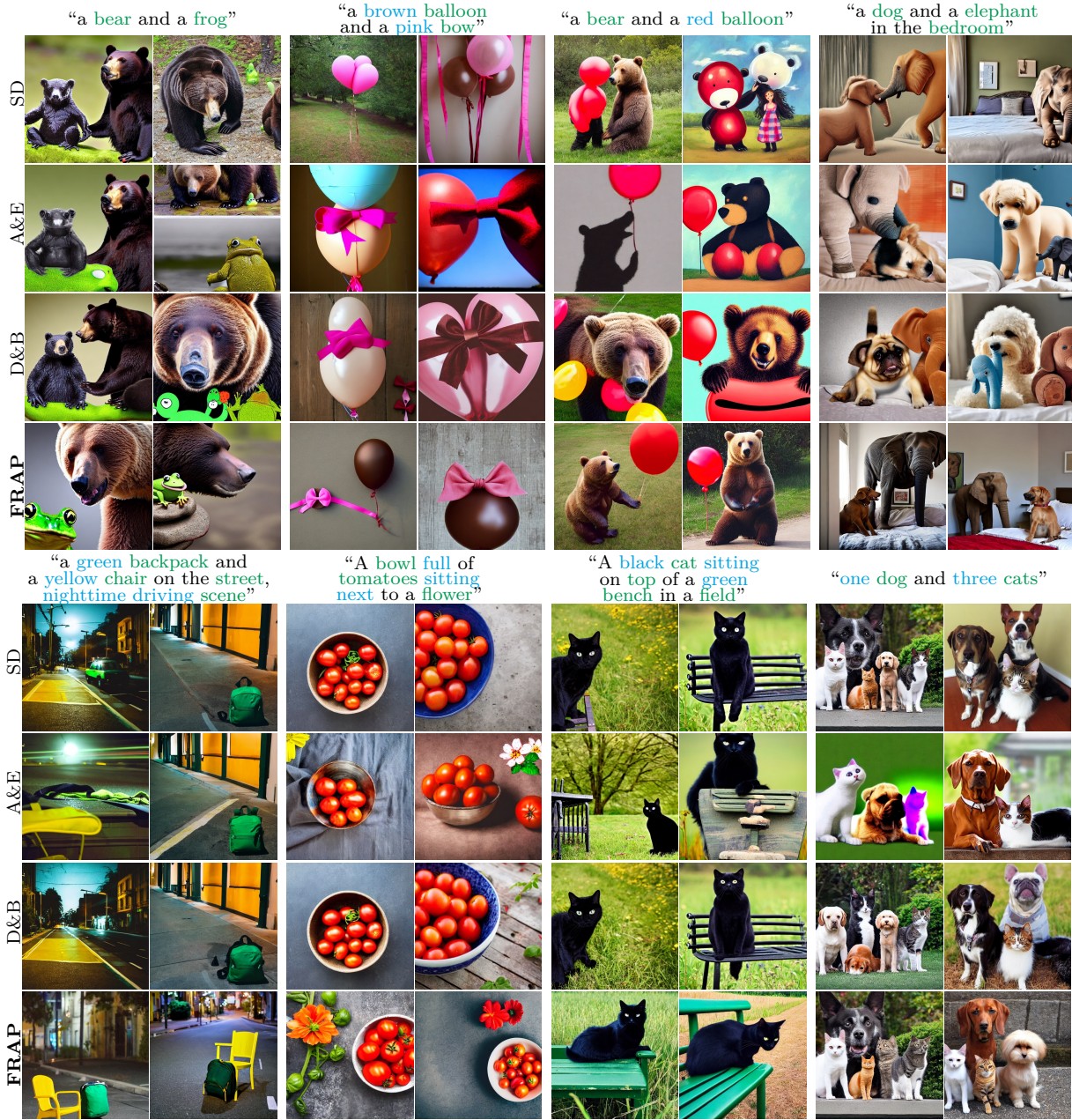

Figure 4: **Qualitative comparison on prompts from different datasets (more in the appendix).** For each prompt, we show images generated by all four methods (using the same set of seeds). The object tokens are highlighted in green and modifier tokens are highlighted in blue.

with complex and challenging prompts, while remaining on par with these methods on simple datasets with prompts created from templates. Moreover, our generated images have significantly higher authenticity with more realistic appearances as confirmed by the CLIP-IQA-Real metric and visual comparisons. Although D&B can achieve on-par or higher performance in image quality reflected by CLIP-IQA and HPSv2 metrics on some datasets, it consistently has significantly lower performance than FRAP in CLIP-IQA-Real metric, which shows that our method generates images with better authenticity and a more realistic appearance. Meanwhile, FRAP has lower average latency and lower number of UNet calls than these methods as we do not rely on costly refinement loops which repeatedly call UNet at each time-step, especially on datasets

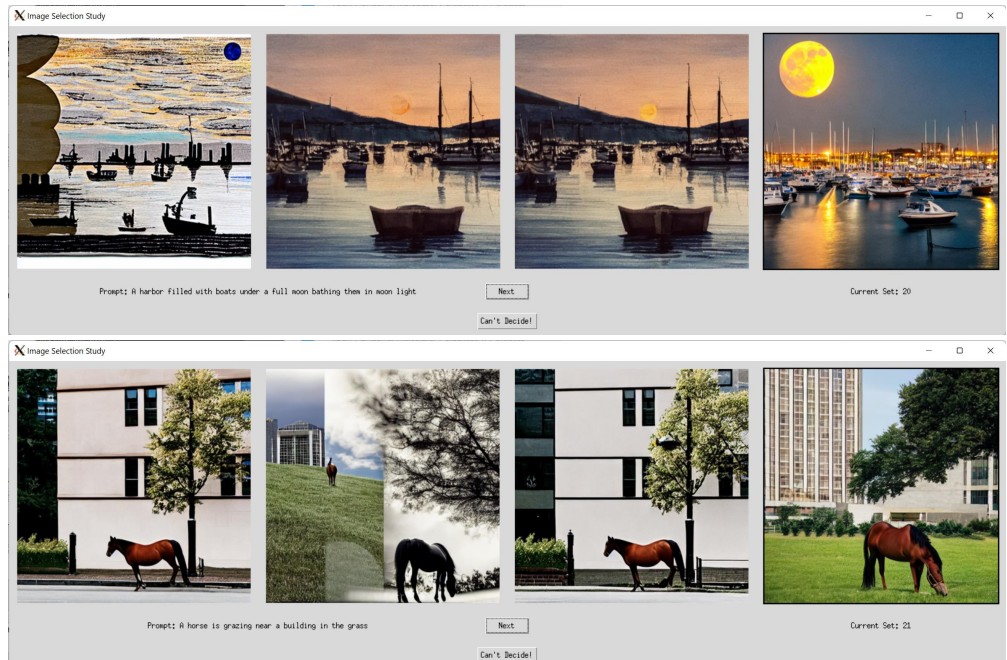

Figure 5: **Human evaluation interface.** See detailed results in Sec. 5.5.

with more complex prompts, e.g., on average 4 seconds faster than D&B for generating one image on the COCO-Subject dataset.

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

## Appendix

This appendix to our main paper has the following sections:

- In App. A, we provide the algorithm overview, implementation details, and experimental settings.

- In App. B, we present more details of our quantitative results and additional qualitative (visual) comparison examples.

- In App. C, we present an ablation study on the design of FRAP.

- In App. D, we present additional quantitative evaluations on additional baseline methods, additional larger evaluation datasets, and applying FRAP to the SDXL model.

- In App. E, we discuss the observed limitations and failing cases.

- In App. F, we provide visualizations and analyses of the loss values, denoising process, and cross-attention maps when using FRAP.

## A  Implementation and Evaluation Details

### A.1  Algorithm Overview

In Algorithm 1, we provide the detailed algorithm of our proposed FRAP method.

---

**Algorithm 1** Algorithm Overview of FRAP

**Input:** Conditional text embedding $c^y$, unconditional text embedding $c^\varnothing$, trained $SD$ model, decoder $\mathcal{D}$
**Output:** The generated image $\mathcal{I}$
Selection of Latent Code:

1: $z_T^b \sim \mathcal{N}(\mathbf{0}, \mathbf{I}), \forall b \in \{1, 2, ..., B\}$
2: $Z_T \leftarrow [z_T^1, z_T^2, ..., z_T^B]$                                         ▷ Batch of Initial Latent Codes
3: **for** $t = T, T-1, ..., t_{\text{select}}$ **do**
4:     $Z_{t-1}, [A_t^1, A_t^2, ..., A_t^B] \leftarrow SD(Z_t, t, c^y, c^\varnothing)$
5: **end for**
6: $b^* \leftarrow \text{argmin}_{b \in \{1, 2, ..., B\}} \mathcal{L}(A_{t_{\text{select}}}^b)$
Adaptive Prompt Weighting:

1: $z_T \leftarrow z_T^{b*}, \alpha_T \leftarrow \vec{0}$
2: **for** $t = T, T-1, ..., t_{\text{end}}$ **do**                                       ▷ Adaptive Prompt Weighting
3:     $\phi_t \leftarrow \phi_{LB} + (\phi_{UB} - \phi_{LB})\sigma(\alpha_t)$
4:     $\tilde{c}^y \leftarrow \phi_t \odot c^y + (1 - \phi_t) \odot c^\varnothing$
5:     $z_{t-1}, A_t \leftarrow SD(z_t, t, \tilde{c}^y, c^\varnothing)$
6:     $\alpha_{t-1} \leftarrow \alpha_t - \eta_t \cdot \nabla_{\alpha_t} \mathcal{L}(A_t)$
7: **end for**
8: **for** $t = t_{\text{end}} - 1, ..., 0$ **do**                                        ▷ No Prompt Weighting
9:     $z_{t-1} \leftarrow SD(z_t, t, c^y, c^\varnothing)$
10: **end for**
11: **Return** $\mathcal{I} \leftarrow \mathcal{D}(z_0)$

---

### A.2  Implementation Details

Following Chefer et al. (2023); Li et al. (2023), we use the $16 \times 16$ CA units for computing the objective function. The weight of object-modifier binding loss $\lambda = 1$. For the optimization in Eq. (9), we use a constant step-size $\eta_t = \eta = 1$ (see ablation in Table 12). Chefer et al. (2023) observed that the final time-steps do

not alter the spatial locations/structures of objects. As a result, we apply our adaptive prompt weighting method to a subset of time-steps $t = T, T-1, ..., t_{\text{end}}$, where $T = 50$ and $t_{\text{end}} = 26$ (see ablation in Table 10).[7]

For our adaptive prompt weighting method, the token weighting coefficients $\phi$ for the special $\langle sot \rangle$, $\langle eot \rangle$ and *padding* tokens are set to 1 and are frozen (i.e., not updated). Similar to the suggested prompt weighting values of Hertz et al. (2022), we set the lower and upper bound of per-token weighting coefficients to $\phi \in [\phi_{LB} = 0.6, \phi_{UB} = 1.4]$. For selecting the initial latent code, we perform 15 steps of inference from $t = T = 50$ to $t_{\text{select}} = 36$ with a batch of $|B| = 4$ noisy latent codes sampled from $\mathcal{N}(\mathbf{0}, \mathbf{I})$. Following the other hyperparameters used by Chefer et al. (2023), we use CFG guidance scale $\beta = 7.5$ and the Gaussian filter kernel size is 3 with a standard deviation of 0.5.

## A.3 Experimental Settings

### A.3.1 Baselines

We compare our approach with three baseline methods. (i) Stable Diffusion (SD) (Rombach et al., 2022), (ii) Attend-and-Excite (A&E) (Chefer et al., 2023), and (iii) Divide&Bind (D&B) (Li et al., 2023). For the A&E (Chefer et al., 2023) and D&B (Li et al., 2023) baselines, we follow their default settings in the papers and implementations. All baselines and our approach are evaluated based on the Stable Diffusion 1.5 (Rombach et al., 2022; Hugging Face, 2023b) model following the choice of D&B (Li et al., 2023) at FP16 precision with its default PNDM (Liu et al., 2022) scheduler. Our additional experiments show that FRAP is also applicable to SDXL (Podell et al., 2023) and can also improve its performance (see Table 15).

### A.3.2 Object and Modifier Identification

To automate the manual identification of object tokens and modifier tokens, we used the spaCy's transformer-based dependency parser (Honnibal & Montani, 2017) (also used by SynGen (Chefer et al., 2023)) to automatically extract a set of object tokens $S$ and a set of modifier tokens $R$. For each object token $s$, we use $R_s$ to represent its related modifier tokens $r \in R_s$. We use $(s, r)$ to represent an object-modifier pair (e.g. "brown dog"), where $P$ is the pairing operation and $P(s, R_s)$ represents the set of all object-modifier pairs between object token $s$ and its related modifier tokens $r \in R_s$. We use $P(S, R)$ to represents the set of all object-modifier pairs identified in the prompt. To guarantee the correctness of automatically extracted nouns and modifiers, humans can manually check the predictions and make any corrections if necessary.

### A.3.3 Datasets

The recent methods A&E (Chefer et al., 2023) and D&B (Li et al., 2023) introduce new prompt datasets for evaluating the effectiveness of a T2I model in handling catastrophic neglect and modifier binding. In this work, we adopt the same datasets and use them for the evaluation of all the baselines. For all datasets used in the experiments of the main paper, we generate 64 images for each prompt using the same set of 64 seeds for each evaluated method.

In Table 4, we utilize the dependency parser (Honnibal & Montani, 2017) to analyze the average number of tokens, average number of object tokens, and average number of relation pairs for each dataset to indicate the complexity of each dataset. The datasets below are sorted based on the average number of tokens per prompt in each dataset (i.e., from lowest complexity to highest complexity):

1. **Animal-Animal** (Chefer et al., 2023), which handles the simple cases with two animal objects in the prompt (evaluates the *catastrophic neglect* issue).

2. **Multi-Object** (Li et al., 2023), which aims to present multiple object entities in the prompt with a count.

---

[7]Note that we are overloading the term "time-step" here. What we mean by *time-step*, going forward, is the *jump index*. That is, going from $t = 50$ to $t = 49$, for example, we are in fact jumping from actual time-step **1000** to $1000 - \frac{1000}{50} = \mathbf{980}$. We could decide to perform inference in fewer [or more] number of jumps, however, we use the default value of $T = 50$ to remain comparable to the literature.

3. **Animal-Object** (Chefer et al., 2023), which has prompts with an animal and a colored object.

4. **Color-Object** (Chefer et al., 2023), which assigns a color to each of the two objects (evaluates the *modifier binding* issue).

5. **Animal-Scene** (Li et al., 2023), which contains a scene or scenario in the prompt, along with two animals.

6. **COCO-Attribute** (Li et al., 2023), which focuses on the modifier-related prompts filtered from MSCOCO (Lin et al., 2014) captions used in Hu et al. (2023) with up to two objects and with diverse modifiers.

7. **COCO-Subject** (Li et al., 2023), which focuses on the objects-related prompts filtered from MSCOCO (Lin et al., 2014) captions used in Hu et al. (2023) with up to four objects and without modifiers.

8. **Color-Object-Scene** (Li et al., 2023), which introduces a scene or scenario along with the two colored objects.

9. **COCO-5K**. We adopt the validation set of the MS-COCO dataset (Lin et al., 2014) which contains the ground-truth reference image. To select prompt-image pairs that are most relevant to prompt-image alignment evaluation and specifically object presence and object-modifier binding, we only keep the relevant prompts with at least two objects and each has at least one modifier word by filtering the prompts with the spaCy (Honnibal & Montani, 2017) language dependency parser. This filtering process selects 16k most relevant prompts from the original 40k prompts in MS-COCO, and we randomly sample a 5k subset from the 16k most relevant prompts. We refer to this dataset as COCO-5K and will release this dataset to facilitate reproducibility and further research.

Table 4: **Description and statistics of datasets**. The order of the datasets is sorted based on the average number of tokens per prompt in each dataset which indicates the complexity of the dataset.

| Dataset | Description | Number of Prompts | Average Number of (i.e., Per Prompt) | | |
|---|---|---|---|---|---|
| | | | Tokens | Object Tokens | Relation Pairs |
| Animal-Animal (Chefer et al., 2023) | a [animalA] and a [animalB] | 66 | 5.0 | 2.0 | 0.0 |
| Multi-Object (Li et al., 2023) | more than two objects or instances in the image | 30 | 5.4 | 2.0 | 2.1 |
| Animal-Object (Chefer et al., 2023) | a [animal] and a [color][object] | 144 | 5.8 | 2.0 | 0.8 |
| Color-Object (Chefer et al., 2023) | a [colorA][subjectA] and a [colorB][subjectB] | 66 | 7.0 | 2.0 | 2.0 |
| Animal-Scene (Li et al., 2023) | a [animalA] and a [animalB] [scene] | 56 | 9.7 | 4.0 | 0.9 |
| COCO-Attribute (Li et al., 2023) | filtered COCO captions related to the attributes | 27 | 10.3 | 2.9 | 2.9 |
| COCO-Subject (Li et al., 2023) | filtered COCO captions related to subjects in the image | 30 | 10.9 | 4.0 | 1.7 |
| Color-Obj-Scene (Li et al., 2023) | a [colorA][subjectA] and a [colorB][subjectB][scene] | 60 | 12.0 | 4.2 | 3.1 |
| COCO-5K | prompts with at least two objects and each has at least one modifier | 5000 | 12.1 | 4.2 | 3.1 |

### A.3.4 Evaluation Metrics

We use multiple metrics to evaluate the performance of various methods in three different aspects: prompt-image alignment, overall image quality, and image authenticity.

To evaluate the prompt-image alignment, we consider the following metrics in our experimental evaluation:

- **Text-to-Text Similarity (TTS) (Chefer et al., 2023; Li et al., 2023).** TTS computes the cosine similarity between the embeddings of (i) original text prompt and (ii) caption of the synthesized image generated with a captioning model such as BLIP (Li et al., 2022).

- **Full-prompt Similarity (Full-CLIP) (Chefer et al., 2023).** This metric computes the average cosine CLIP (Radford et al., 2021) similarity between the prompt text embedding and generated image embedding.

- **Minimum Object Similarity (MOS) (Chefer et al., 2023).** Paiss et al. (2022) observed that only using the Full-CLIP to estimate the prompt-image alignment may not be completely reliable, as a high Full-CLIP score is still achievable even when a few objects are missing in the generated image. MOS measures the minimum cosine similarity between the image embedding and text embeddings of each sub-prompt containing an object mentioned in the prompt.

To evaluate the overall image quality, we consider the following Image Quality Assessment (IQA) metrics in our experimental evaluation:

- **Human Preference Score v2 (HPSv2) (Wu et al., 2023).** HPSv2 metric reflects the human preference so it evaluates the image authenticity, quality, and faithfulness to the prompt. It is a scoring mechanism which is built on top of CLIP, by fine-tuning on Human-Preference Dataset v2 (HPDv2) (Wu et al., 2023) to accurately predict the human preferences for synthesized images.

- **CLIP-IQA (Wang et al., 2023).** This metric assesses both the quality perception (i.e., the general look of the images: e.g., overall quality, brightness, etc.) and abstract perception (i.e., the general feel of the image: e.g., aesthetic, happy, etc.). Specifically, we consider the averaged CLIP-IQA score on four aspects including ("quality", "noisiness", "natural", and "real") to evaluate the overall image quality of the generated image.

To evaluate the image authenticity, we utilize the CLIP-IQA-Real score on the "real" aspect of the CLIP-IQA metric (Wang et al., 2023) which specifically evaluates the level of image authenticity and how realistic the generated images are.

### A.3.5 Latency

Our reported latency measures the average wall-clock time for generating one image on each dataset in seconds with a V100 GPU.

## B  Detailed Results

### B.1  Detailed Quantitative Results

For the experiments in the main paper, we present the detailed results on all datasets in Table 5.

We observe in Table 5 that, on datasets with *simple* hand-crafted templates (Animal-Animal, Color-Object, and Animal-Object), there is a tight competition among A&E, D&B, and FRAP across all metrics, while all three exceed the performance of vanilla SD. This basically shows that the *simple* datasets may not be sufficient to definitively identify the superior method among the competing options.

We see a clear advantage of FRAP on more challenging *complex* datasets. In Table 5, we observe that FRAP consistently outperforms all other methods in almost all metrics on the *complex* datasets. Specifically, FRAP

Table 5: **Quantitative comparison** of FRAP and other methods across different datasets. Higher is better (except for latency). Our reported latency measures the average wall-clock time for generating one image on each dataset in seconds with a V100 GPU. For the experiments in this table, we generate 64 images for each prompt using the same set of 64 seeds for each method.

| Dataset | Method | Prompt-Image Alignment | | | Overall Image Quality | | Image Authenticity | Latency (s) ↓ |
|---|---|---|---|---|---|---|---|---|
| | | TTS ↑ | Full-CLIP ↑ | MOS ↑ | HPS v2 ↑ | CLIP-IQA ↑ | CLIP-IQA Real ↑ | |
| Animal-Animal (2023) | SD (2022) | 0.763 | 0.312 | 0.217 | 0.268 | 0.822 | 0.937 | **8.1** |
| | A&E (2023) | **0.810** | **0.334** | **0.250** | 0.272 | 0.822 | 0.937 | 12.1 |
| | D&B (2023) | 0.802 | 0.330 | 0.242 | **0.273** | **0.836** | 0.952 | 12.9 |
| | FRAP (ours) | 0.804 | 0.333 | 0.245 | 0.272 | 0.817 | **0.962** | 13.1 |
| Multi-Object (2023) | SD (2022) | 0.763 | 0.304 | 0.244 | 0.264 | 0.792 | 0.811 | **8.0** |
| | A&E (2023) | **0.793** | **0.315** | **0.263** | 0.266 | 0.792 | 0.811 | 12.3 |
| | D&B (2023) | 0.789 | 0.314 | 0.258 | **0.270** | **0.816** | 0.824 | 13.7 |
| | FRAP (ours) | 0.783 | 0.311 | 0.256 | 0.266 | 0.804 | **0.869** | 13.2 |
| Animal-Object (2023) | SD (2022) | 0.791 | 0.343 | 0.247 | 0.277 | 0.756 | 0.826 | **7.8** |
| | A&E (2023) | 0.833 | 0.357 | **0.267** | 0.283 | 0.755 | 0.824 | 11.6 |
| | D&B (2023) | 0.831 | 0.351 | 0.261 | **0.284** | **0.770** | 0.817 | 12.5 |
| | FRAP (ours) | **0.835** | **0.361** | **0.267** | 0.283 | 0.763 | **0.874** | 13.2 |
| Color-Object (2023) | SD (2022) | 0.761 | 0.338 | 0.236 | 0.281 | 0.617 | 0.482 | **7.9** |
| | A&E (2023) | **0.811** | **0.363** | **0.270** | **0.286** | 0.617 | 0.482 | 12.1 |
| | D&B (2023) | 0.804 | 0.357 | 0.261 | **0.286** | 0.637 | 0.519 | 14.9 |
| | FRAP (ours) | 0.802 | 0.357 | 0.259 | **0.286** | **0.653** | **0.566** | 13.6 |
| Animal-Scene (2023) | SD (2022) | 0.736 | 0.347 | 0.229 | 0.273 | 0.777 | 0.928 | **8.0** |
| | A&E (2023) | 0.741 | 0.354 | **0.243** | 0.275 | 0.777 | 0.928 | 17.3 |
| | D&B (2023) | 0.755 | 0.356 | 0.239 | **0.276** | **0.789** | 0.963 | 16.8 |
| | FRAP (ours) | **0.758** | **0.360** | 0.236 | **0.276** | 0.779 | **0.964** | 13.8 |
| COCO-Attribute (2023) | SD (2022) | 0.817 | 0.347 | 0.245 | 0.286 | 0.703 | 0.750 | **8.1** |
| | A&E (2023) | 0.820 | 0.353 | **0.254** | 0.286 | 0.703 | 0.750 | 15.3 |
| | D&B (2023) | 0.821 | 0.353 | 0.251 | 0.288 | 0.720 | 0.760 | 16.7 |
| | FRAP (ours) | **0.838** | **0.358** | **0.254** | **0.289** | **0.725** | **0.778** | 13.9 |
| COCO-Subject (2023) | SD (2022) | 0.829 | 0.331 | 0.250 | 0.278 | 0.805 | 0.852 | **7.9** |
| | A&E (2023) | 0.829 | 0.333 | 0.254 | 0.277 | 0.805 | 0.852 | 19.1 |
| | D&B (2023) | 0.835 | 0.333 | 0.253 | **0.279** | **0.814** | 0.862 | 18.1 |
| | FRAP (ours) | **0.837** | **0.334** | **0.255** | **0.279** | **0.814** | **0.866** | 14.0 |
| Color-Obj-Scene (2023) | SD (2022) | 0.707 | 0.366 | 0.254 | 0.282 | 0.675 | 0.770 | **8.0** |
| | A&E (2023) | 0.708 | 0.376 | 0.267 | 0.286 | 0.675 | 0.770 | 17.9 |
| | D&B (2023) | 0.719 | 0.375 | 0.264 | 0.286 | **0.682** | 0.808 | 16.7 |
| | FRAP (ours) | **0.727** | **0.381** | **0.269** | **0.287** | 0.676 | **0.835** | 14.3 |
| COCO-5K | SD (2022) | 0.801 | 0.322 | 0.247 | 0.276 | 0.712 | 0.784 | **4.2** |
| | A&E (2023) | 0.799 | 0.324 | 0.250 | 0.275 | 0.705 | 0.782 | 21.3 |
| | D&B (2023) | 0.802 | 0.323 | 0.248 | 0.276 | 0.713 | 0.785 | 18.2 |
| | FRAP (ours) | **0.811** | **0.327** | **0.251** | **0.279** | **0.750** | **0.849** | 11.2 |

is consistently best on *complex* datasets in terms of TTS, Full-CLIP, HPSv2, and CLIP-IQA-Real metrics, which demonstrates the ability of FRAP to generate realistic images in various complicated scenarios. FRAP surpasses its competitors in all evaluation metrics on the COCO-Attribute and COCO-Subject datasets, which contain more realistic real-world prompts compared to manually crafted datasets.

FRAP performs well on most evaluation datasets, except for the Multi-Object dataset, where all methods struggle to generate the correct number of dogs and cats (e.g., see the bottom-right of Fig. 4). In general, delivering a correct count of objects in generated images is still an open challenge. See App. E for discussion of the limitations.

### B.2   Additional Qualitative Results

In Figs. 13 and 14, we provide additional visual comparisons among FRAP and other methods on prompts from different datasets. As shown in the figures, FRAP generates images that are more faithful to the prompt containing all the mentioned objects without hybrid objects and have accurate modifier bindings. Moreover, images generated with FRAP maintain a more realistic appearance.

## C   Ablation Study

In this section, we present ablation studies for different design aspects of FRAP. For all ablation experiments, we generate 16 images for each prompt using the same set of 16 seeds for each evaluated variant of FRAP. We report the same prompt-image alignment metrics and image quality assessment metrics used in the main paper and provide details on the metrics in App. A.3.

### C.1   Objective Function

First, we ablate the design of our unified objective function $\mathcal{L}$. In Table 6, we remove the binding loss $\mathcal{L}_{\text{binding}}$ in the second row and remove the presence loss $\mathcal{L}_{\text{presence}}$ in the third row. We observe that removing the binding loss has a smaller impact while removing the presence loss has a significant drop in performance. It shows the important role of object presence loss in improving both prompt-image alignment and image quality since the presence of each object is fundamental to the faithfulness of the image. Overall, using both loss terms achieves the highest performance. Therefore, we use both loss terms for our approach by default.

In the fourth row of Table 6, we attempt to use only the loss from the most neglected token with the lowest max activation. This max variant, similar to the loss in A&E (Chefer et al., 2023), results in lower performance than our default approach. The higher performance shows the advantage of our presence loss design which uses the *mean* among all objects as the total object presence loss and enhances the presence of all objects instead of a single one, which is essential for complex prompts with multiple objects.

Table 6: **Ablation study on the binding loss $\mathcal{L}_{\text{binding}}$, presence loss $\mathcal{L}_{\text{presence}}$, and max aggregation of per-object presence losses $\mathcal{L}^s_{\text{presence}}$.** The default setting using both $\mathcal{L}_{\text{binding}}$ and $\mathcal{L}_{\text{presence}}$ achieves the overall best performance in both prompt-image alignment and image quality. We generate 16 images for each prompt in the COCO-Attribute (Li et al., 2023) dataset using the same set of 16 seeds for each evaluated variant.

| Method | Prompt-Image Alignment | | | Overall Image Quality | | Image Authenticity | Latency (s) ↓ |
|---|---|---|---|---|---|---|---|
| | TTS ↑ | Full-CLIP ↑ | MOS ↑ | HPS v2 ↑ | CLIP-IQA ↑ | CLIP-IQA Real ↑ | |
| FRAP (default) | 0.836 | **0.359** | **0.256** | **0.289** | **0.722** | **0.773** | 14.1 |
| w/o $\mathcal{L}_{\text{binding}}$ | **0.837** | **0.359** | 0.255 | **0.289** | 0.720 | 0.772 | 14.1 |
| w/o $\mathcal{L}_{\text{presence}}$ | 0.807 | 0.341 | 0.239 | 0.284 | 0.707 | 0.756 | 13.9 |
| max $\mathcal{L}^s_{\text{presence}}$ | 0.827 | 0.352 | 0.252 | 0.288 | 0.712 | 0.752 | 14.1 |

Furthermore, to better understand the internal workings of the method, we visualize the cross-attention (CA) maps while using only the presence loss, only the binding loss, or both loss terms. In Fig. 6, with only $\mathcal{L}_{\text{presence}}$, objects are separated but color green leaks to the crown. With only $\mathcal{L}_{\text{binding}}$, the color binding is correct but two objects are overlapped. Overall, using both loss components achieves the best result.

### C.2   Object Presence Loss

In Table 7, we ablate the choice of our object presence loss. We replace the object presence loss with the Total Variation loss (Li et al., 2023). Our proposed object presence loss in Eq. (5) performs better than the TV loss.

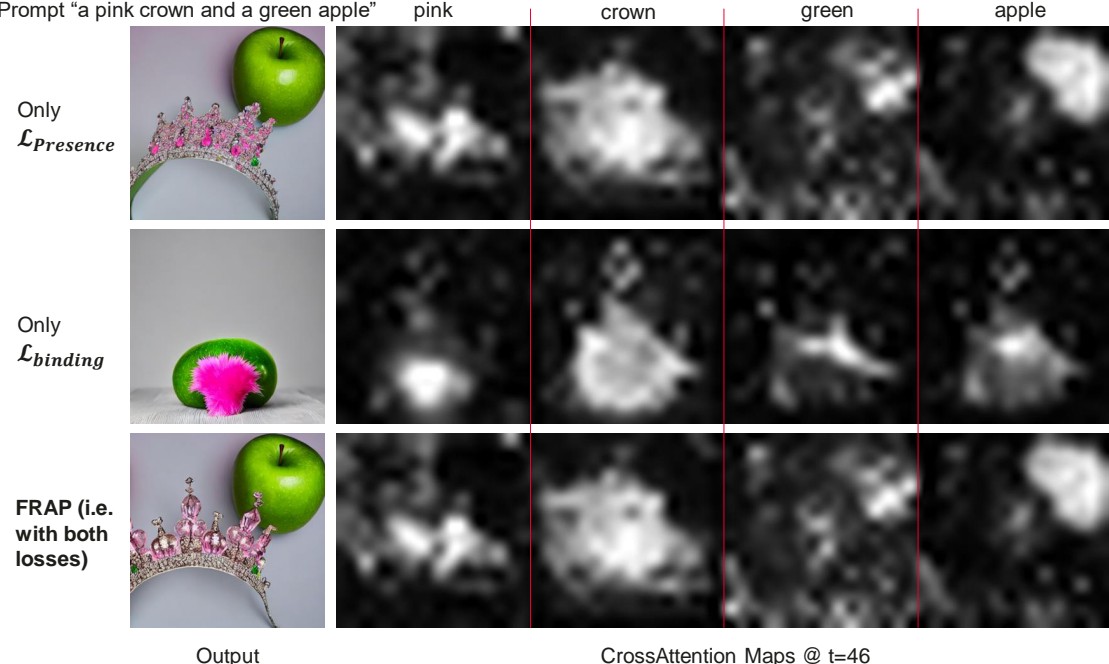

Figure 6: **Visualization of cross-attention maps** while using different loss function components.

Table 7: **Quantitative comparison of our object presence loss $\mathcal{L}_{\text{presence}}$ and the Total Variation (TV) loss (Li et al., 2023).** Our object presence loss $\mathcal{L}_{\text{presence}}$ achieves better performance in both prompt-image alignment and image quality. We generate 16 images for each prompt in the Color-Obj-Scene (Li et al., 2023) dataset using the same set of 16 seeds for each loss.

| Method | Prompt-Image Alignment | | | Overall Image Quality | | Image Authenticity | Latency (s) ↓ |
|---|---|---|---|---|---|---|---|
| | TTS ↑ | Full-CLIP ↑ | MOS ↑ | HPS v2 ↑ | CLIP-IQA ↑ | CLIP-IQA Real ↑ | |
| FRAP ($\mathcal{L}_{\text{presence}}$) | **0.725** | **0.380** | **0.269** | **0.287** | **0.670** | **0.829** | 13.9 |
| FRAP (TV) | 0.718 | 0.375 | 0.263 | 0.286 | 0.664 | 0.806 | 14.0 |

### C.3 Object-Modifier Binding Loss

In Table 8, we compare our object-modifier binding loss to other alternatives. Our proposed object-modifier binding loss in Eq. (7) improves the performance over the JSD (Li et al., 2023) and KLD (Rassin et al., 2023) loss. Our approach of considering the minimum probability essentially filters out the outliers, since the outliers in two distributions rarely overlap, minimizing their impact on our objective. This is unlike KLD and JSD which consider the entire distributions and therefore, can be disproportionately inflated by outliers.

### C.4 Adaptive Prompt Weighting

In Table 9, we present an ablation on the effect of Adaptive Prompt Weighting (APW) and the Initial Latent Code Selection (ILCS) process. We evaluate a Static Prompt Weighting (SPW) baseline, where we set the prompt token weighting coefficients to a fixed value of 1.4 for the object and attribute tokens in the first 25 steps aiming to improve the prompt-image alignment. As shown in the second row, using SPW improves prompt-image alignment but significantly degrades image quality and image authenticity compared to vanilla SD (Rombach et al., 2022) in the first row reflected by the lower CLIP-IQA and CLIP-IQA-Real scores. In the third row, unlike SPW, our proposed APW improves prompt-image alignment while maintaining or

Table 8: **Quantitative comparison of our object-modifier binding loss $\mathcal{L}_{\mathbf{binding}}$ and the JSD loss (Li et al., 2023) and KLD loss (Rassin et al., 2023).** Our object-modifier binding loss $\mathcal{L}_{\text{binding}}$ achieves better performance in both prompt-image alignment and image quality. We generate 16 images for each prompt in the Color-Obj-Scene (Li et al., 2023) dataset using the same set of 16 seeds for each loss.

| Method | Prompt-Image Alignment | | | Overall Image Quality | | Image Authenticity | Latency (s) ↓ |
|---|---|---|---|---|---|---|---|
| | TTS ↑ | Full-CLIP ↑ | MOS ↑ | HPS v2 ↑ | CLIP-IQA ↑ | CLIP-IQA Real ↑ | |
| FRAP ($\mathcal{L}_{\text{binding}}$) | **0.725** | **0.380** | **0.269** | **0.287** | **0.670** | 0.829 | 13.9 |
| FRAP (JSD) | 0.721 | **0.380** | 0.266 | 0.286 | 0.662 | 0.825 | 14.1 |
| FRAP (KLD) | 0.724 | **0.380** | 0.267 | **0.287** | 0.668 | **0.831** | 14.2 |

even improving image quality compared to vanilla SD (Rombach et al., 2022). It reflects that our APW can preserve image quality by *adaptively* setting the weighting coefficients, while SPW degrades the image quality due to using a *constant* value throughout the generation and requiring humans to select its values manually.

Overall, FRAP which uses both our proposed APW method and the ILCS process results in the best comprehensive performance. In the third row of Table 9, using only APW results in lower prompt-image alignment scores and image quality scores compared to FRAP which combines both APW and ILCS. In the fourth row of Table 9, using only ILCS improves both prompt-image alignment and image quality compared to vanilla SD. Although using only ILCS results in a higher score in the CLIP-IQA Real metric than FRAP, the prompt-image alignment performance in all three metrics degrades compared to FRAP which shows the necessity of using APW for improving prompt-image alignment. In the fifth row, we try to combine SPW and ILCS, which results in overall lower prompt-image alignment and lower image quality compared to FRAP, which uses adaptive prompt weighting instead of a static one.

Table 9: **Ablation study on adaptive prompt weighting**. FRAP using both Adaptive Prompt Weighting (APW) and Initial Latent Code Selection (ILCS) achieves the overall best performance in both prompt-image alignment and image quality. We generate 16 images for each prompt in COCO-Attribute (Li et al., 2023) dataset using the same set of 16 seeds for each evaluated variant of FRAP.

| Method | Prompt-Image Alignment | | | Overall Image Quality | | Image Authenticity | Latency (s) ↓ |
|---|---|---|---|---|---|---|---|
| | TTS ↑ | Full-CLIP ↑ | MOS ↑ | HPS v2 ↑ | CLIP-IQA ↑ | CLIP-IQA Real ↑ | |
| SD (Rombach et al., 2022) | 0.815 | 0.345 | 0.245 | 0.287 | 0.710 | 0.746 | 7.7 |
| Static Prompt Weighting (SPW) | 0.841 | 0.356 | 0.249 | 0.288 | 0.692 | 0.694 | 8.0 |
| Adaptive Prompt Weighting (APW) | 0.829 | 0.354 | 0.250 | 0.287 | 0.718 | 0.746 | 10.5 |
| Initial Latent Code Selection (ILCS) | 0.828 | 0.354 | 0.253 | 0.288 | **0.723** | **0.797** | 13.1 |
| SPW and ILCS | **0.844** | 0.358 | 0.250 | 0.288 | 0.689 | 0.697 | 13.5 |
| FRAP (i.e. APW and ILCS) | 0.836 | **0.359** | **0.256** | **0.289** | 0.722 | 0.773 | 14.1 |

For qualitative examples of this ablation, see Fig. 7. Without selecting a decent initial latent code through the ILCS process, the first row of images has an "incorrect layout", e.g. "dog", "shoe", and "red" take the majority of the space and the other objects are neglected. In the second row, using the ILCS process alone without APW can find a "good layout" to generate both objects, but the prompt-image alignment is suboptimal in image details compared to FRAP in the third row using both components (see improved details in the dog's eye and the car).

### C.5 Time-Steps to Perform Adaptive Prompt Weighting

In Table 10, we compare different numbers of time-steps to perform adaptive prompt weighting. By default, we use $t_{\text{end}} = 26$ so the adaptive prompt weighting is applied to the first 25 steps in the reverse generative process (RGP). By using adaptive prompt weighting only in the first 10 steps in the RGP with $t_{\text{end}} = 41$, there is a slight drop in prompt-image alignment and a slight increase in IQA metric scores. When using

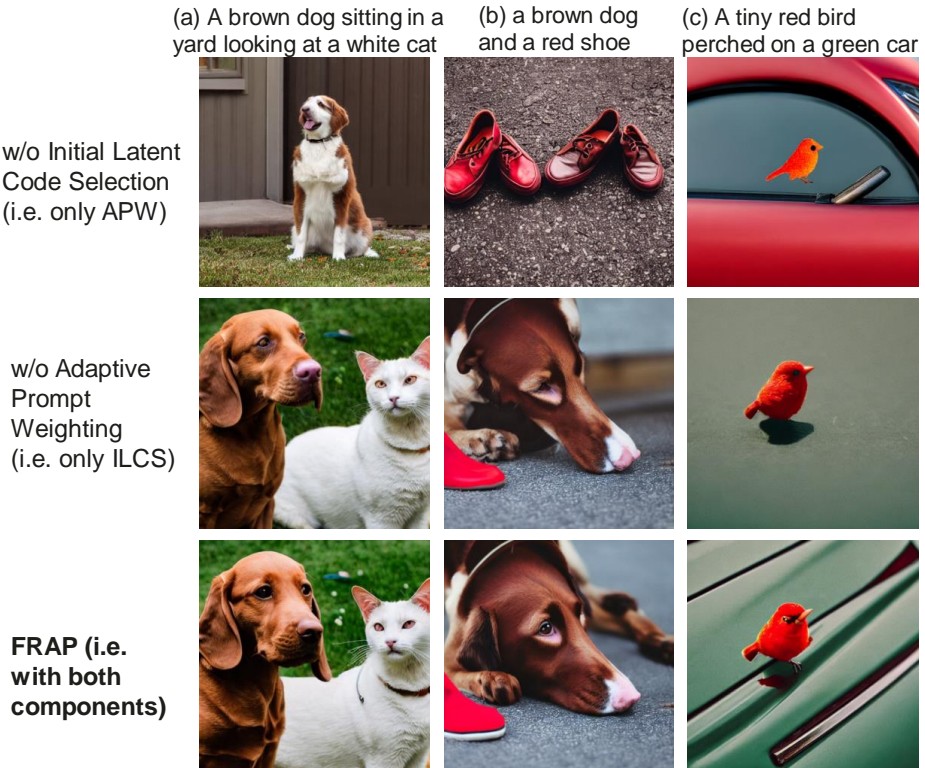

Figure 7: **Qualitative comparisons** for the ablation on Adaptive Prompt Weighting (APW) and Initial Latent Code Selection (ILCS).

adaptive prompt weighting in all 51 RGP steps with $t_{\text{end}} = 0$, the latency increases with no significant benefit in performance. This aligns with the observation of Chefer et al. (2023) that the final time-steps do not alter the spatial locations/structures of objects. Therefore, we use $t_{\text{end}} = 26$ to achieve a good balance between performance and latency.

Table 10: **Ablation study on the number of time-steps to apply adaptive prompt weighting.** We generate 16 images for each prompt in the COCO-Attribute (Li et al., 2023) dataset using the same set of 16 seeds for each evaluated variant of FRAP.

| Method | Prompt-Image Alignment | | | Overall Image Quality | | Image Authenticity | Latency (s) ↓ |
|---|---|---|---|---|---|---|---|
| | TTS ↑ | Full-CLIP ↑ | MOS ↑ | HPS v2 ↑ | CLIP-IQA ↑ | CLIP-IQA Real ↑ | |
| FRAP (default) | 0.836 | 0.359 | **0.256** | **0.289** | 0.722 | 0.773 | 14.1 |
| $t_{\text{end}} = 41$ | 0.832 | 0.355 | 0.254 | **0.289** | **0.726** | **0.787** | 13.6 |
| $t_{\text{end}} = 0$ | 0.837 | **0.360** | **0.256** | **0.289** | 0.718 | 0.769 | 15.3 |

## C.6 Number of Optimization Steps

In Table 11, we perform an ablation study on the *number of optimization steps* (# of Opti. Steps) to perform at each time-step, comparing a range of values from 1 to 6. The number of optimization steps means how many times to repeat the inference of the current time-step $t$ with an updated $\alpha$ at each time-step. FRAP uses only 1 optimization step at each time-step since it directly uses the updated $\alpha$ in the next time-step $t - 1$. For the variants with 2 to 6 optimization steps, we repeat the current time-step $t$ inference for "# of

Opti. Steps" times with an updated $\alpha$, which will increase the inference latency while having a possibility for improvements in prompt-image alignment.

From the results in Table 11, we observe that the setting used in this paper with 1 optimization step achieves the best performance in image authenticity and overall image quality, while having lower latency and comparable prompt-image alignment. The lower latency is due to only performing a single UNet call at each time-step, instead of more repeated UNet calls that are required when repeating the inference of each time-step for running the optimization steps.

Table 11: **Ablation study on the number of optimization steps** (# of Opti. Steps) to perform at each time-step. We generate 16 images for each prompt in the COCO-Attribute (Li et al., 2023) dataset using the same set of 16 seeds for each evaluated variant of FRAP.

| Method | Prompt-Image Alignment | | | Overall Image Quality | | Image Authenticity | Latency (s) ↓ |
|---|---|---|---|---|---|---|---|
| | TTS ↑ | Full-CLIP ↑ | MOS ↑ | HPS v2 ↑ | CLIP-IQA ↑ | CLIP-IQA Real ↑ | |
| # of Opti. Steps = 1 (this paper) | 0.836 | **0.359** | **0.256** | 0.289 | **0.722** | **0.773** | 14.1 |
| # of Opti. Steps = 2 | 0.840 | **0.359** | **0.256** | 0.289 | 0.716 | 0.766 | 16.1 |
| # of Opti. Steps = 3 | **0.843** | 0.358 | 0.253 | 0.289 | 0.715 | 0.758 | 20.8 |
| # of Opti. Steps = 4 | 0.841 | **0.359** | 0.252 | 0.289 | 0.712 | 0.751 | 23.6 |
| # of Opti. Steps = 5 | 0.842 | **0.359** | 0.253 | 0.289 | 0.710 | 0.745 | 27.2 |
| # of Opti. Steps = 6 | 0.840 | **0.359** | 0.252 | 0.289 | 0.710 | 0.742 | 30.5 |

## C.7  Step-Size

In Table 12, we perform an ablation study on the step-size $\eta$. By comparing results on a range of $\eta$ values from 0.2 to 5.0, we observe that increasing $\eta$ improves the prompt-image alignment, whereas decreasing $\eta$ favors the image quality and image authenticity. The setting of $\eta = 1$ used in this paper achieves a good balance between prompt-image alignment, overall image quality, and image authenticity.

Table 12: **Ablation study on the step-size** $\eta$ (i.e., learning rate) used when updating $\alpha$ in Eq. (10). We generate 16 images for each prompt in the COCO-Attribute (Li et al., 2023) dataset using the same set of 16 seeds for each evaluated variant of FRAP.

| Method | Prompt-Image Alignment | | | Overall Image Quality | | Image Authenticity | Latency (s) ↓ |
|---|---|---|---|---|---|---|---|
| | TTS ↑ | Full-CLIP ↑ | MOS ↑ | HPS v2 ↑ | CLIP-IQA ↑ | CLIP-IQA Real ↑ | |
| $\eta = 0.2$ | 0.833 | 0.355 | 0.252 | 0.289 | **0.728** | **0.793** | 14.1 |
| $\eta = 0.5$ | 0.835 | 0.356 | 0.253 | 0.289 | 0.724 | 0.782 | 14.2 |
| $\eta = 1.0$ (default) | 0.836 | **0.359** | **0.256** | 0.289 | 0.722 | 0.773 | 14.1 |
| $\eta = 2.0$ | **0.839** | 0.358 | 0.254 | 0.289 | 0.718 | 0.767 | 14.0 |
| $\eta = 5.0$ | 0.835 | 0.357 | 0.253 | 0.289 | 0.715 | 0.752 | 14.1 |

# D  Additional Quantitative Evaluations

## D.1  Additional Baselines

In Table 13, we compare FRAP with two additional text-to-image methods SynGen (Rassin et al., 2023) and StructureDiffusion (Feng et al., 2022). Although SynGen and StructureDiffusion have competitive CLIP-IQA scores with similar latency to vanilla SD, these two approaches fall short in the prompt-image alignment scores. Overall, FRAP has consistently higher prompt-image alignment scores and is also competitive in the IQA metrics.

Table 13: Quantitative comparison of FRAP with **two additional text-to-image methods Syn-Gen (2023) and StructureDiffusion (2022)**. For the experiments in this table, we generate 16 images for each prompt using the same set of 16 seeds for each method.

| Dataset | Method | Prompt-Image Alignment | | | Overall Image Quality | | Image Authenticity | Latency (s) ↓ |
|---|---|---|---|---|---|---|---|---|
| | | TTS ↑ | Full-CLIP ↑ | MOS ↑ | HPS v2 ↑ | CLIP-IQA ↑ | CLIP-IQA Real ↑ | |
| COCO-Subject (2023) | SD (2022) | 0.829 | 0.330 | 0.249 | 0.278 | 0.810 | 0.863 | 8.2 |
| | SynGen (2023) | 0.827 | 0.334 | **0.255** | **0.279** | **0.819** | 0.863 | 6.4 |
| | Structure-Diffusion (2022) | 0.827 | 0.331 | 0.249 | 0.277 | 0.811 | **0.884** | 6.0 |
| | A&E (2023) | 0.829 | 0.333 | 0.252 | 0.277 | 0.801 | 0.843 | 19.1 |
| | D&B (2023) | 0.835 | 0.333 | 0.252 | **0.279** | 0.812 | 0.855 | 17.7 |
| | FRAP (ours) | **0.839** | **0.335** | **0.255** | **0.279** | 0.812 | 0.871 | 13.3 |
| COCO-Attribute (2023) | SD (2022) | 0.815 | 0.345 | 0.245 | 0.287 | 0.710 | 0.746 | 7.7 |
| | SynGen (2023) | 0.818 | 0.352 | 0.250 | 0.287 | **0.736** | **0.774** | 7.6 |
| | Structure-Diffusion (2022) | 0.822 | 0.349 | 0.246 | 0.285 | 0.712 | 0.767 | 6.0 |
| | A&E (2023) | 0.813 | 0.354 | 0.255 | 0.286 | 0.697 | 0.731 | 14.9 |
| | D&B (2023) | 0.822 | 0.351 | 0.250 | 0.288 | 0.716 | 0.750 | 16.5 |
| | FRAP (ours) | **0.836** | **0.359** | **0.256** | **0.289** | 0.722 | 0.773 | 14.1 |

## D.2 Additional Datasets

In Table 14, we further evaluate FRAP and the other methods on the DrawBench (Saharia et al., 2022) dataset and the ABC-6K (Feng et al., 2022) dataset in addition to the datasets listed in Table 4. For the DrawBench (Saharia et al., 2022) dataset, we use a total of 183 comprehensive prompts excluding 7 prompts from the "Rare Words" category and 10 prompts from the "Misspellings" category since these are not within the scope of this paper or the other considered baselines. The ABC-6K (Feng et al., 2022) dataset contains 6.4K prompts filtered from MSCOCO (Lin et al., 2014) where each contains at least two color words modifying different objects. For the DrawBench dataset, we generate 16 images for each prompt using the same set of 16 seeds for each method. For the ABC-6K dataset, we generate one image for each prompt using the same seed for each method.

Table 14: Quantitative comparison of FRAP and other methods on **two additional datasets Draw-Bench (2022) and ABC-6K (2022)**. For the DrawBench dataset, we generate 16 images for each prompt using the same set of 16 seeds for each method. For the ABC-6K dataset, we generate one image for each prompt using the same seed for each method.

| Dataset | Method | Prompt-Image Alignment | | | Overall Image Quality | | Image Authenticity | Latency (s) ↓ |
|---|---|---|---|---|---|---|---|---|
| | | TTS ↑ | Full-CLIP ↑ | MOS ↑ | HPS v2 ↑ | CLIP-IQA ↑ | CLIP-IQA Real ↑ | |
| DrawBench (2022) | SD (2022) | 0.713 | 0.329 | 0.243 | 0.275 | 0.703 | 0.820 | **8.1** |
| | A&E (2023) | 0.718 | **0.336** | **0.251** | 0.275 | 0.710 | 0.826 | 18.8 |
| | D&B (2023) | 0.720 | 0.334 | 0.248 | 0.276 | **0.722** | 0.832 | 17.6 |
| | FRAP (ours) | **0.722** | **0.336** | 0.248 | **0.277** | 0.717 | **0.837** | 14.5 |
| ABC-6K (2022) | SD (2022) | 0.771 | 0.332 | 0.249 | 0.282 | 0.675 | 0.698 | **7.9** |
| | A&E (2023) | 0.767 | 0.334 | 0.251 | 0.281 | 0.675 | 0.697 | 19.3 |
| | D&B (2023) | 0.771 | 0.333 | 0.250 | 0.283 | 0.688 | 0.703 | 18.3 |
| | FRAP (ours) | **0.783** | **0.338** | **0.253** | **0.284** | **0.726** | **0.788** | 14.2 |

As shown in Table 14, FRAP outperforms the other methods in terms of four out of six evaluated metrics while having a lower latency than A&E (Chefer et al., 2023) and D&B (Li et al., 2023) on the DrawBench

(Saharia et al., 2022) dataset. Moreover, FRAP significantly outperforms the other methods in all metrics in terms of prompt-image alignment and image quality on the large-scale ABC-6K (Feng et al., 2022) dataset.

### D.3 SDXL Model

From visual (qualitative) examples in Figs. 8, 9, and 10, we observe that the vanilla SDXL suffer from prompt-image alignment issues such as catastrophic neglect and incorrect modifier binding similar to the vanilla SD1.5 (Rombach et al., 2022) model.

Therefore, we apply FRAP to the SDXL (Podell et al., 2023) model and show that FRAP is generalizable to different T2I diffusion models. Specifically, in Figs. 8, 9, and 10, we observe significant improvements in image quality, image realness, prompt-image alignment, and the counting ability when applying FRAP to SDXL, including but not limited to the examples provided below:

**Image realness and image quality.** For example, in Fig. 8, for the prompt "a black cat and a red suitcase on the street, snowy driving scene", the image generated by FRAP (2nd column, 4th row) is much more realistic and the overall image quality is also better compared to the cartoonish image generated by vanilla SDXL (2nd column, 3rd row). The same applies to prompts "a blue bird ..." and "a purple dog ..." in Fig. 8. Similarly, this can also be seen in the images for prompts "a laptop ...", "a man ...", and "a woman sitting on ..." in Fig. 9.

**Prompt-image alignment.** For example, in Fig. 8, for the prompt "a green backpack and a yellow chair in the library", the image generated by FRAP (4th column, 4th row) is much better aligned with the prompt (i.e., have both "backpack" and "chair" generated in the correct color), compared to the image generated by vanilla SDXL (4th column, 3rd row) which does not have a backpack, same applies to prompts "a pink clock ..." and "a green balloon ..." in Fig. 8. Similarly, the better alignment of FRAP can also be seen in the images for prompts "a dog and a cat ...", "a little dog jumping ...", and "a woman rides her bike ..." in Fig. 9.

**Counting ability.** In Fig. 10 (for the Multi-Object dataset), we frame the image in red if it has an incorrect object count. It can be seen that FRAP (12 out of 16 images are correct) generates images with a correct number of objects more often than vanilla SDXL (only 2 out of 16 images are correct). Fewer red frames are shown for FRAP.

With the extensive qualitative evaluation on SDXL (Podell et al., 2023), we indeed observe gains from FRAP in realness, quality, alignment, and counting. These extensive qualitative examples together with all the quantitative metrics in Table 15 verify that the performance benefit of FRAP is convincing, robust, and is not only limited to SD1.5 (Rombach et al., 2022) model.

Table 15: Additional quantitative results on **applying FRAP to the SDXL (2023) model**. For the experiments in this table, we generate 16 images for each prompt using the same set of 16 seeds for each method. See visual (qualitative) comparisons in Figs. 8, 9, and 10.

| Dataset | Method | Prompt-Image Alignment | | | Overall Image Quality | | Image Authenticity | Latency (s) ↓ |
|---|---|---|---|---|---|---|---|---|
| | | TTS ↑ | Full-CLIP ↑ | MOS ↑ | HPS v2 ↑ | CLIP-IQA ↑ | CLIP-IQA Real ↑ | |
| Color-Obj-Scene (Li et al., 2023) | SDXL (2023) | 0.742 | 0.388 | 0.267 | 0.293 | **0.791** | 0.940 | 27.2 |
| | FRAP (with SDXL) | **0.744** | **0.391** | **0.272** | **0.295** | 0.789 | **0.943** | 44.7 |
| COCO-Subject (Li et al., 2023) | SDXL (2023) | 0.844 | 0.346 | **0.257** | 0.283 | 0.825 | 0.830 | 29.0 |
| | FRAP (with SDXL) | **0.845** | **0.347** | **0.257** | **0.284** | **0.832** | **0.836** | 45.6 |
| Multi-Object (Li et al., 2023) | SDXL (2023) | 0.785 | 0.307 | 0.249 | 0.264 | 0.838 | **0.903** | 28.7 |
| | FRAP (with SDXL) | **0.787** | **0.311** | **0.253** | **0.266** | **0.842** | 0.901 | 44.1 |

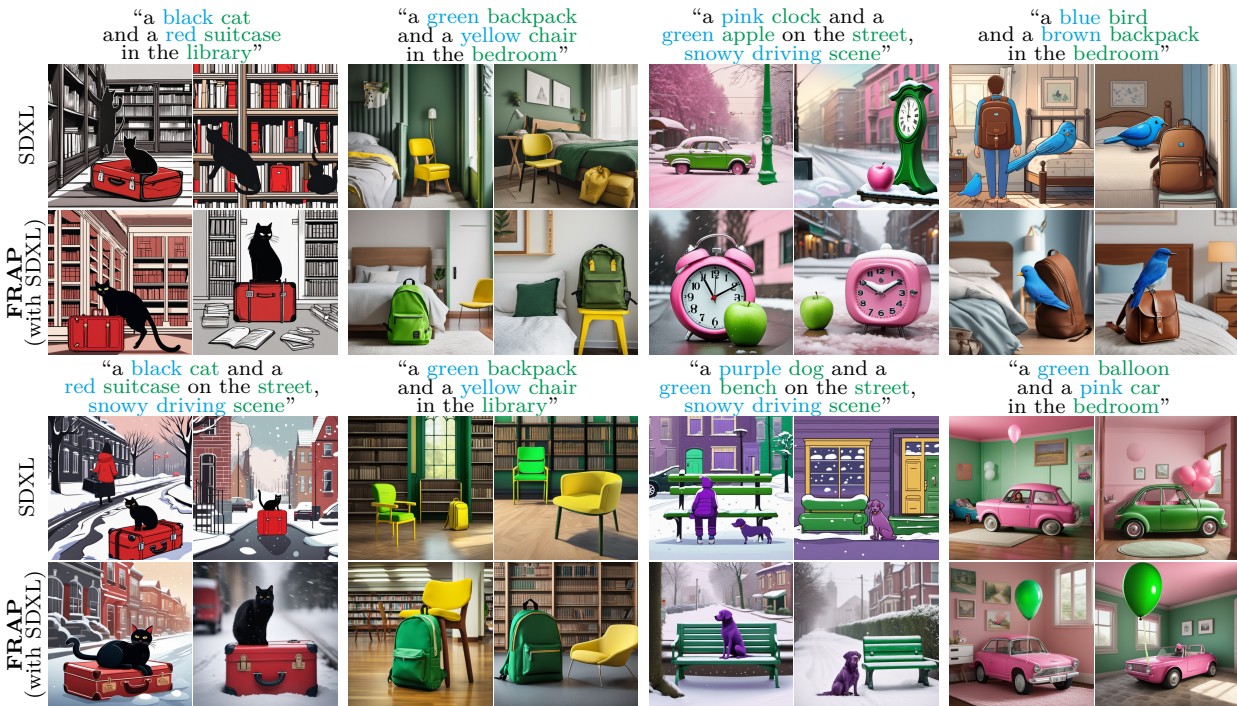

Figure 8: **Qualitative comparison of SDXL and FRAP (with SDXL) on Color-Obj-Scene (Li et al., 2023).** The same set of seeds is used for both methods. The object tokens are highlighted in green and modifier tokens are highlighted in blue. Quantitative results are in Table 15.

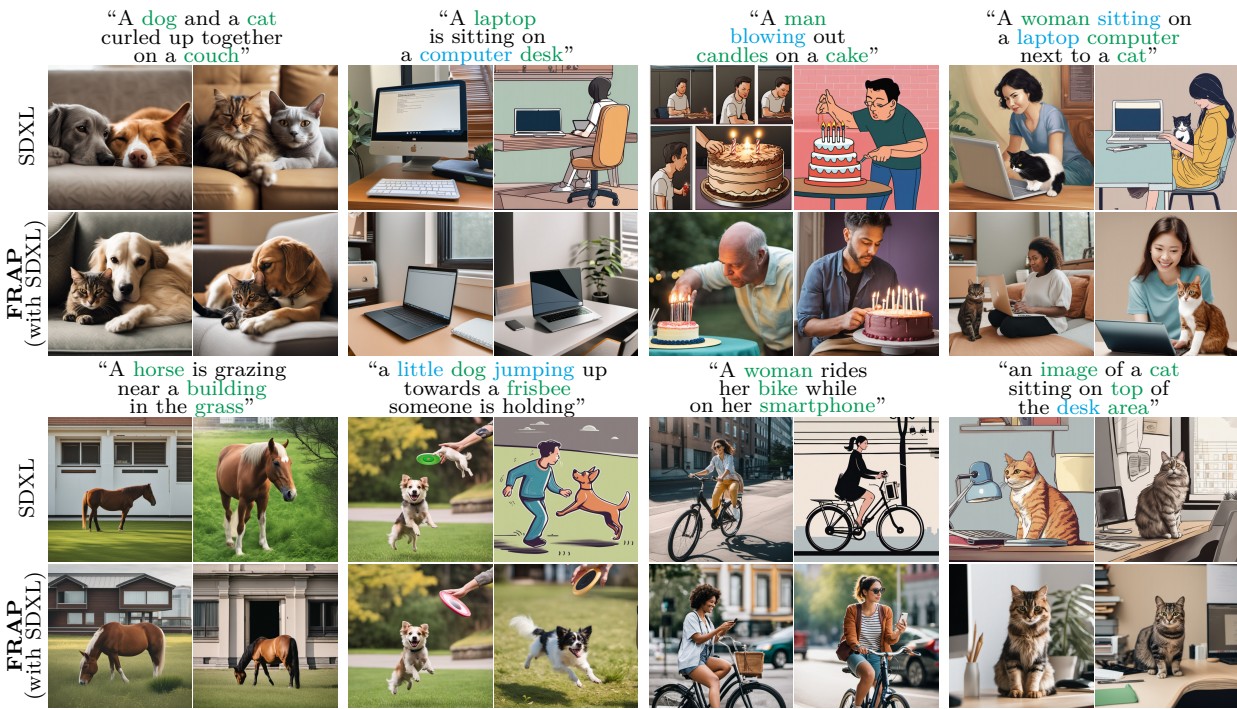

Figure 9: **Qualitative comparison of SDXL and FRAP (with SDXL) on COCO-Subject (Li et al., 2023).** The same set of seeds is used for both methods. The object tokens are highlighted in green and modifier tokens are highlighted in blue. Quantitative results are in Table 15.

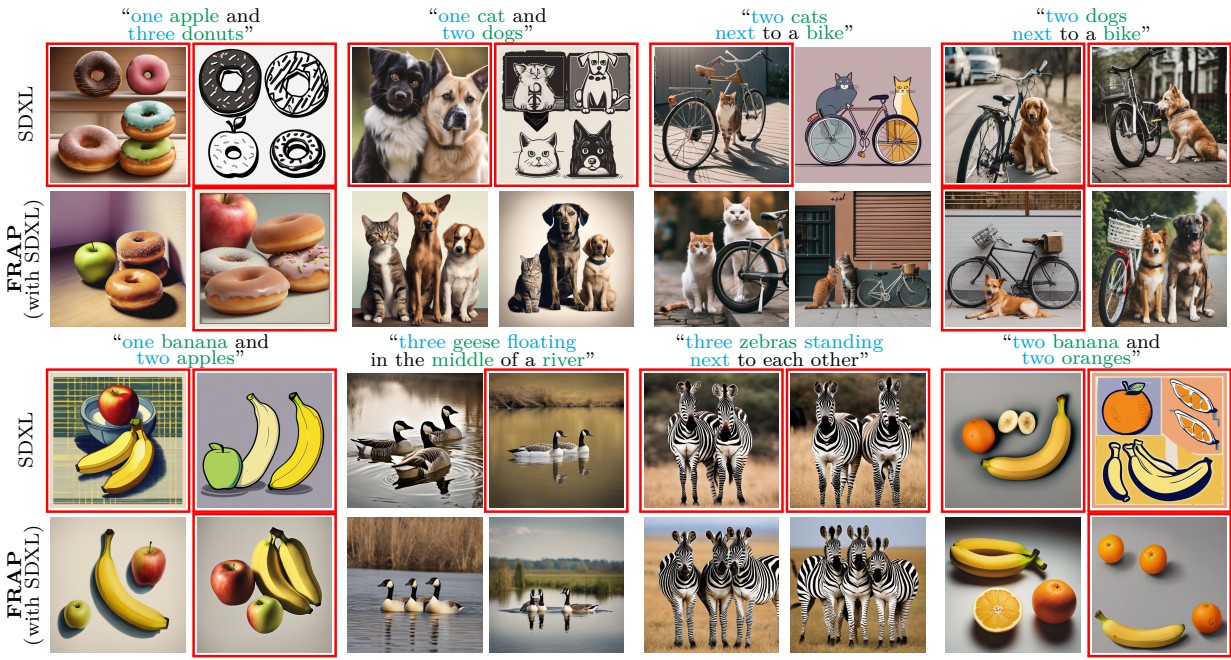

Figure 10: **Qualitative comparison of SDXL and FRAP (with SDXL) on Multi-Object (Li et al., 2023).** The same set of seeds is used for both methods. The object tokens are highlighted in green and modifier tokens are highlighted in blue. Quantitative results are in Table 15. We frame the generated images with an incorrect object count in red color.

# E   Limitations and Failing Cases

We observe in Table 5 that FRAP performs well on most evaluation datasets, except for the Multi-Object dataset, where FRAP and all other methods including Standard SD (Rombach et al., 2022), A&E (Chefer et al., 2023), and D&B (Li et al., 2023) struggle to generate the correct number of dogs and cats (e.g., see the failing cases in bottom-right of Fig. 4). Similarly, the vanilla SDXL (Podell et al., 2023) model also struggles to generate the correct number of objects as shown in Fig. 10, where we frame the image in red if it has an incorrect object count. In Fig. 10, it can be seen that FRAP (12 out of 16 images are correct) generates images with a correct number of objects more often than vanilla SDXL (only 2 out of 16 images are correct). Therefore, FRAP can improve SDXL's counting ability as seen in fewer red frames shown for FRAP. In general, delivering a correct count of objects in generated images is still an open challenge. Our work focuses on a wider scope of achieving superior prompt alignment and image authenticity for complex and general prompts.

To automatically extract objects and modifiers from the prompt, we choose the spaCy (Honnibal & Montani, 2017) parser since it is a lightweight and robust framework dedicated to many standard NLP tasks such as dependency parsing, entity recognition, and sentence segmentation. In this work, we have shown end-to-end performance benefits of FRAP (as a whole stack of inference-time optimization) in an extensive range of settings, while the specific choice of dependency parser is orthogonal to our main technical contribution and focus. That being said, spaCy and any other parsers are not perfect and may occasionally omit a modifier. However, this is not fatal to the generation process of the Diffusion Model (DM), which still relies on the original whole prompt for generation. Specifically, spaCy-identified modifiers are only used to index the cross-attention maps when calculating the binding loss function in the inference-time prompt reweighting optimization. A word in the prompt, even if omitted by spaCy, is still incorporated in the generation process by the DM. For example, for prompt "A dog and a cat curled up together on a couch", spaCy did not extract "curled" as a modifier. However, FRAP can still robustly generate images with the correct semantics of "curled" (see 2nd row, 2nd column in Fig. 9) since the information of "curled" can be passed into the

"cat" and "dog" in the image through the cross-attention mechanism of the original DM. The same applies to the prompt "A laptop is sitting on a computer desk" where the "sitting" modifier is not extracted (see 2nd row, 4th column in Fig. 9). Therefore, FRAP does not critically depend on the extract accuracy of modifier extraction and is robust to failure cases of spaCy.

# F Visualization and Analysis of FRAP

## F.1 Influence of FRAP on the Denoising Process and Attention

To demonstrate the influence of FRAP on the denoising process and how it affects attention, we provide visualizations of the Cross-Attention (CA) maps and the intermediate images every 5 time-steps (see Fig. 11). For both Vanilla SD and FRAP, we show the CA maps for the "horse" token, the CA maps for the "turtle" token, and the intermediate images (i.e., the denoising process) across the time-steps.

At $t = 50$ and $t = 45$, the CA maps for both FRAP and Vanilla SD show that the attention to the "turtle" token is less bright and sparser compared to "horse". By using FRAP, the loss function can detect and give the feedback that "turtle" is receiving less attention (i.e., having a larger loss value). Next, FRAP adaptively adjusts the prompt weighting according to the gradients to emphasize the neglected object "turtle".

At $t = 40$ and $t = 35$, we can clearly observe the influence of FRAP on the denoising process and attention. The "turtle" CA maps for FRAP have a separate blob of high attention values in the bottom left corner, whereas the attention to "turtle" in Vanilla SD is smaller and overlaps with "horse".

Therefore, FRAP can adaptively adjust the prompt weighting to influence the denoising process and attention which improves the prompt-image alignment.

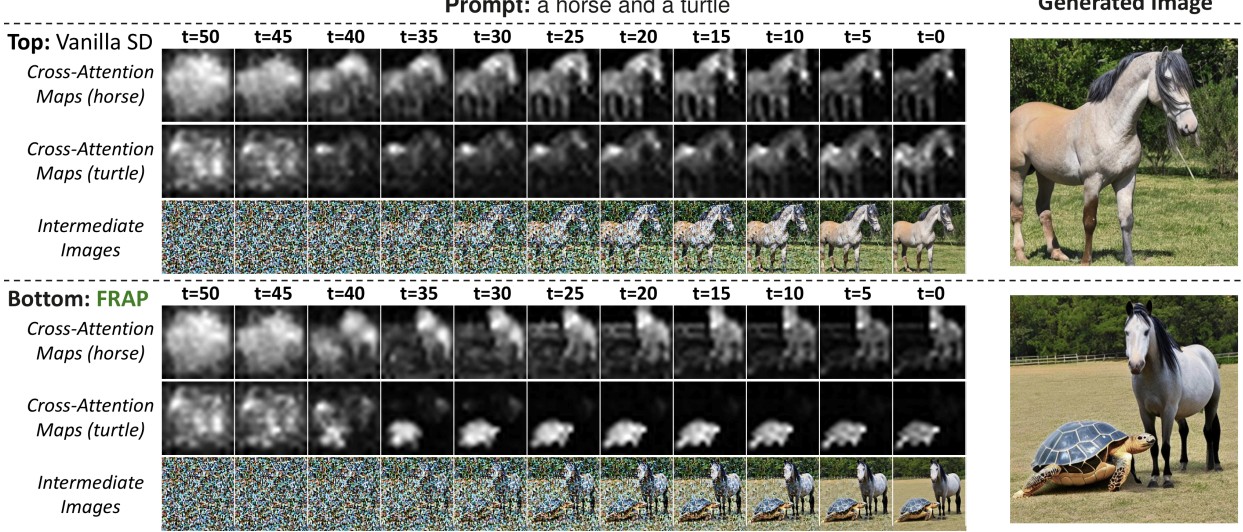

Figure 11: **Visualization of the cross-attention and intermediate images across the time-steps.**

## F.2 Trend of Loss Across Time-Steps

We visualize the trend of the loss across the time-steps during inference. In Fig. 12, we show the generated images (top) and the plots of loss values across the time-steps (bottom), for three different methods: Vanilla SD (left), FRAP with default settings (middle), and FRAP with a larger $\eta$ and larger $t_{end}$ (right).

Referring to Vanilla SD on the left of Fig. 12, we can see that "turtle" is missing from the generated image. This is also reflected by the higher loss value of the orange curve for "turtle" compared to the lower loss value of "horse" represented by the blue curve. In other words, the curves show that Vanilla SD did not give enough attention to "turtle" thus causing it to be neglected in the final generated image.

In contrast, FRAP in the middle of Fig. 12 is able to bring the yellow loss curve for "turtle" down to the same level as the blue curve. During the inference-time optimization, FRAP automatically updates the prompt weighting coefficients based on the feedback gradients from the loss functions. As a result, the model gives more attention to the neglected "turtle" and can generate both objects in the final image.

The other question is to what extent should we minimize the loss, or we should instead aim for good visual quality in the end result (i.e., the generated image). The loss roughly reflects the level of prompt-image alignment, more specifically, how well object presence and object-modifier binding are handled. By minimizing the loss, we are aiming to improve the prompt-image alignment, because the loss function provides feedback signals through the gradients so that FRAP can adaptively update the prompt weighting coefficients in the correct direction of improving prompt-image alignment during inference.

On the right of Fig. 12, we also consider an aggressive variant of FRAP which aims to minimize the loss as much as possible, by increasing the step-size $\eta$ from 1.0 to 5.0, and changing $t_{end}$ to 0 (i.e., applying adaptive prompt weighting in all 51 steps, instead of only in the 25 early steps). Comparing the loss plots of the three methods, we see that this aggressive variant achieves the lowest loss values among the three methods (e.g., the orange curve reaches a very small value of around 0.2). However, the end result of the aggressive variant (right) is much worse than FRAP (middle) despite achieving lower loss values. This can be seen in the quality degradation of the grass background as well as the horse skin, which starts to look like a turtle.

Therefore, we should not aim to minimize the loss as much as possible without considering image quality. Both image visual quality and prompt-image alignment are important for text-to-image generation, so the focus should be finding a good balance between visual quality and minimization of the loss which reflects prompt-image alignment. The step-size $\eta$ controls to what extent the loss gets minimized, as seen in Fig. 12, the loss reaches a much smaller value of around 0.2 (the orange curve in the right plot) when we increase the step-size $\eta$ from 1.0 (middle plot) to 5.0 (right plot). From our ablation experiment results on step-size $\eta$ in Table 12, we can see that increasing the step-size beyond $\eta = 1$ does not increase prompt-image alignment anymore while it reduces image quality. Therefore, with the setting of $\eta = 1$ used in this paper, we achieve a good balance between loss minimization (i.e., prompt-image alignment) and the visual quality of the end result.

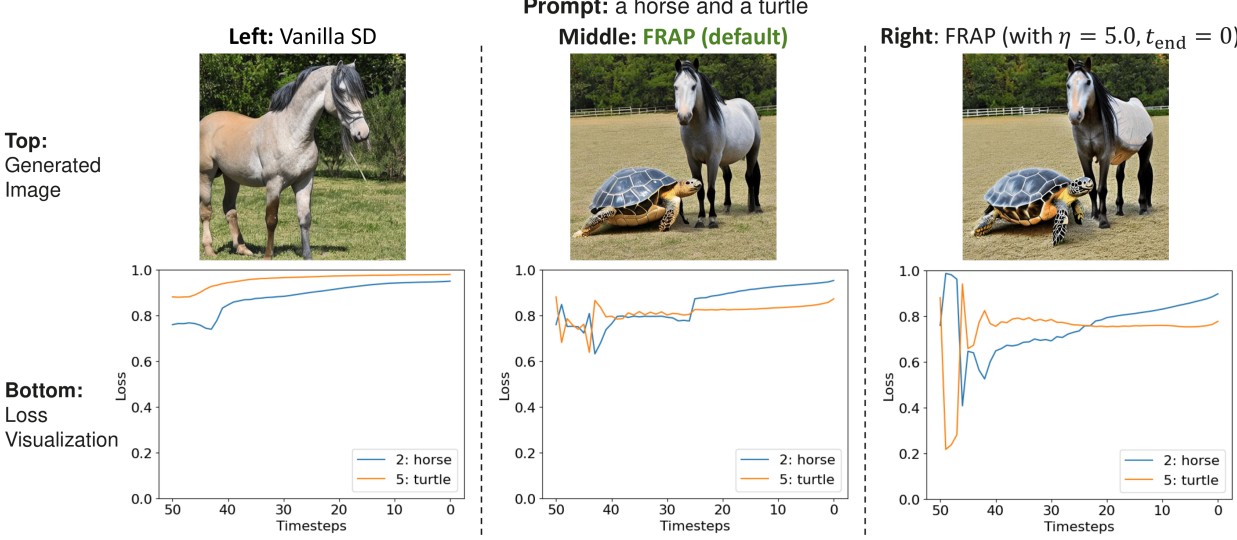

Figure 12: **Visualization of the generated images and plots of loss values across the time-steps.**

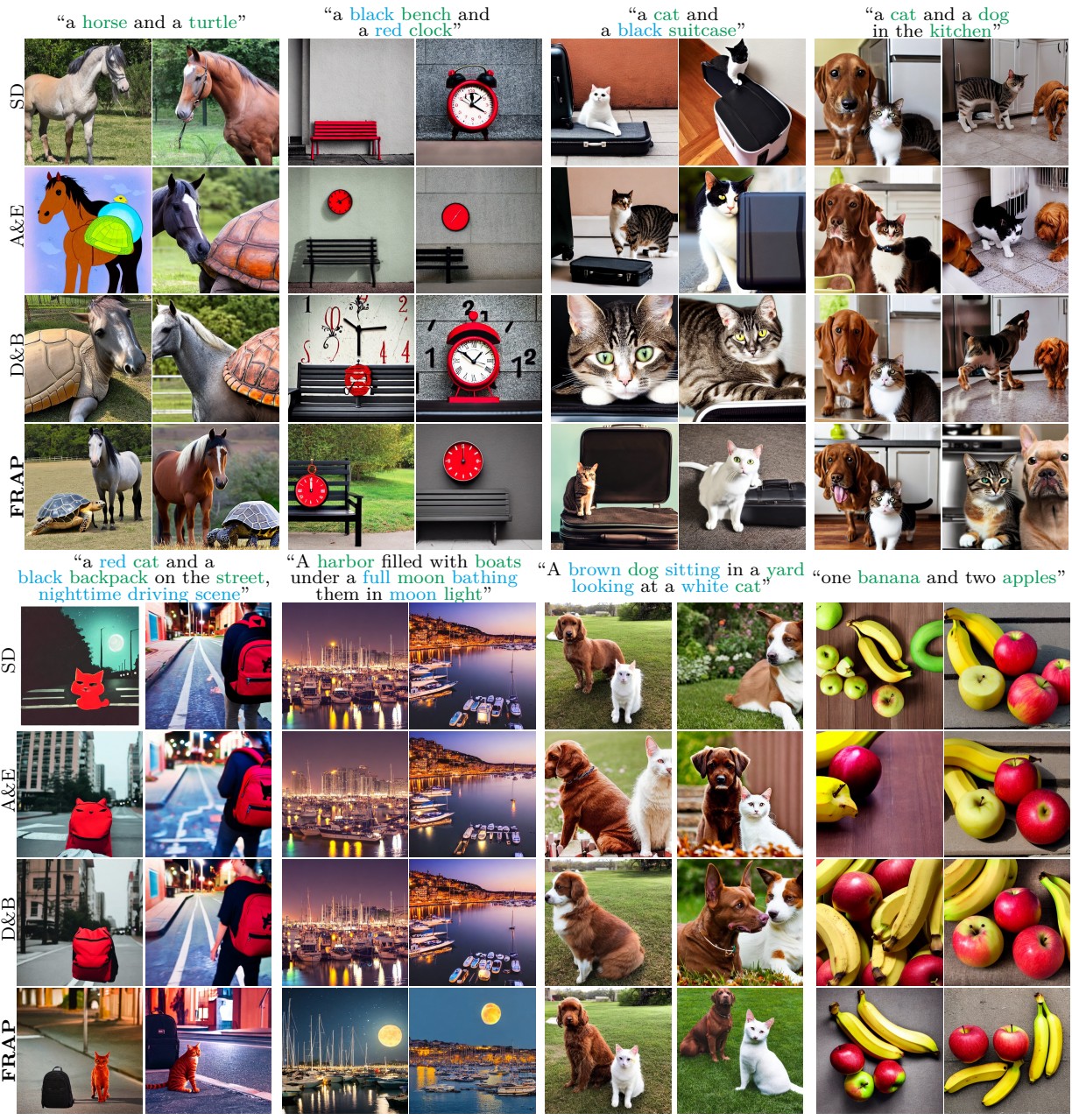

Figure 13: **Qualitative comparison on prompts from different datasets.** For each prompt, we show images generated by all four methods, where we use the same set of seeds. The object tokens are highlighted in green and modifier tokens are highlighted in blue.

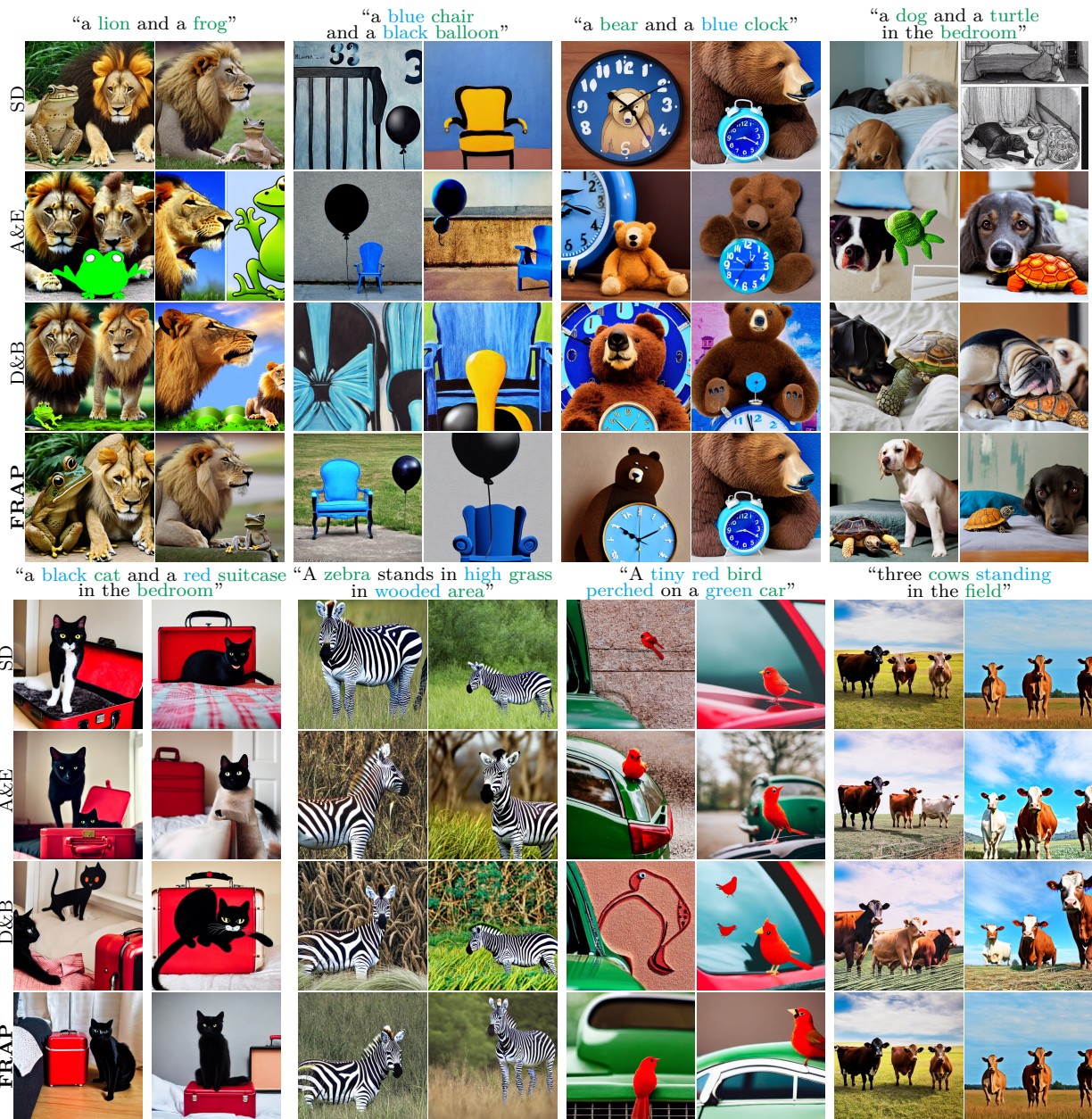

Figure 14: **Qualitative comparison on prompts from different datasets.** For each prompt, we show images generated by all four methods, where we use the same set of seeds. The object tokens are highlighted in green and modifier tokens are highlighted in blue.

