# OpenReview forum: "FRAP: Faithful and Realistic Text-to-Image Generation with Adaptive Prompt Weighting"
_TMLR — Accepted by TMLR_

### Review · Reviewer_fVnT · 2024-12-19

**Summary Of Contributions:**

This paper introduces a novel technique to enhance the faithfulness and realism of text-to-image generation by dynamically adjusting prompt weights during the reverse generative process of diffusion models. The key idea is to optimize a (unified) objective that strengthens object presence and encourages accurate object-modifier binding, ensuring the generated image aligns semantically with the textual prompt.

**Audience:**

Yes

**Broader Impact Concerns:**

No broder impact concerns.

**Claims And Evidence:**

Yes

**Requested Changes:**

This paper presents interesting ideas, but its impact on the research community could be enhanced with a few key revisions.

Primarily, the justification for the proposed method's design choices needs strengthening. While the ablation study (App. C) provides some insights, incorporating those findings and elaborating on the rationale behind specific decisions (especially concerning Eq (5) and Eq (6)) within the main text would greatly improve clarity and reader engagement. This would provide a more compelling narrative for the methodological choices.

Secondly, the quantitative results, while promising, demonstrate only a modest improvement over baseline methods.  The authors should address this directly, discussing the significance of these gains and offering potential explanations for the observed limitations. A deeper analysis of the method's failing cases would provide valuable insights for future research and contribute to a more comprehensive understanding of the method's capabilities.

Finally, to further enhance the evaluation, the authors could consider incorporating more standard metrics for assessing visual quality, such as FID (Fréchet Inception Distance) or IS (Inception Score). This would provide a more comprehensive picture of the method's performance and facilitate comparisons with existing work.

By addressing these points, the authors can significantly strengthen the paper's contribution to the research community and ensure its findings are clearly communicated and understood.

A few other minor typos/comments I found:

- Abstract: “Recent works attempt to improve the faithfulness by optimizing the
latent code, …“ Which latent code?
- Abstract: “FRAP generates images with significantly higher promptimage alignment to prompts from complex datasets,” Writing.
- Abstract: “ faster than D&B” What is D&B? (you haven't introduced this yet)
- Eq(1), space missing in ~
- Eq(1), zT ~ N (0, 1). Should be N(0 Id) (vector instead of scalars)

**Strengths And Weaknesses:**

Strengths:
- The paper proposes a novel and effective method for improving prompt-image alignment in text-to-image generation.
- The method is simple to implement and can be integrated with existing diffusion models.
- The paper provides extensive evaluations on multiple datasets, demonstrating the effectiveness of the proposed method.
- The proposed method outperforms existing methods in terms of both image quality and prompt-image alignment.

Weaknesses:
- The paper proposes a method that appears somewhat arbitrary. While there are numerous ways to address this problem, the authors present only one without much justification for its design choices. Specifically, the design choices made in equations (5) and (6) and the full loss function lack clear motivation or rationale, leaving the reader to question the optimality of the proposed method.
- Some more standard metrics are not reported (FID, IS scores).
- The quantitative results do show some quite modest improvement over the baseline compared methods.
- The paper does not discuss the limitations of the proposed method.

---

> ### Author Response · Authors · 2024-12-28
> **Reply to reviewer fVnT**
>
> We thank the reviewer for reading our manuscript and providing constructive feedback. We updated the manuscript and the changes made are in the violet color (unless specified otherwise in each point below) in the manuscript. The minor typos/comments are fixed. The requested changes by the reviewer are directly addressed in the comments as follows.
>
> **1. Incorporating ablation findings and elaborating on the rationale behind specific decisions (especially concerning Eq (5) and Eq (6)) within the main text.**
>
> We have updated Sec. 4.1.1 and Sec 4.1.2 of the manuscript to elaborate more on the rationale behind design choices of Eq(5) and Eq (6) directly within the main text (Note: the changes made for addressing this point are in blue color).
>
> **2. Observed limitations and failing cases.**
>
> In the Appendix of the manuscript, we added App. E and referenced it in Section 5.1 of the main text to discuss the observed limitations and failing cases. We observe in Table 5 that FRAP performs well on most evaluation datasets, except for the “Multi-Object” dataset, where FRAP and all other evaluated methods, including standard SD (Rombach et al., 2022), A&E (Chefer et al., 2023), and D&B (Li et al., 2023) struggle to generate the correct number of dogs and cats (e.g., see the failing cases in bottom-right of Fig. 4). In general, delivering a correct count of objects in generated images is still an open challenge. Our work focuses on a wider scope of achieving superior image authenticity and prompt alignment for complex and general prompts.
>
> **3. New FID evaluation results, dataset used, and why FID and IS was not originally selected.**
>
> We believe IS (Inception Score) is better suited and more often used for evaluating class-to-image generation task since it was trained for image classification, whereas we focus on the text-to-image generation task. We did not original select the FID (Fréchet Inception Distance) (Heusel et al., 2017) metric since it was not used by the closely related works on this topic (i.e. A&E (Chefer et al., 2023), D&B (Li et al., 2023), StructureDiffusion (Feng et al., 2022)) since it requires a reference ground-truth image, whereas the evaluation datasets adopted by prior works does not contain any reference ground truth image (i.e. Animal-Animal, Color-Object, Animal-Object, Animal-Scene, Color-Objet-Scene, Multi-Object, COCO-Subject, COCO-Attribute datasets). The FID metric is a full-reference IQA metric that requires a ground-truth reference image in addition to the generated image, different from the no-reference IQA metrics selected in our paper (i.e. HPSv2 (Wu et al., 2023) and CLIP-IQA (Wang et al., 2023)) which only requires the generated image.
>
> However, following the reviewer’s suggestion, we evaluated FID during the rebuttal period. We have updated Section 5.1 and added Table 2 to the manuscript to include new FID evaluation results on the COCO-5K dataset (the lower the FID, the better; SD: 39.52; A&E: 35.82, D&B: 39.08; FRAP: 34.22), which shows dramatic advantage of FRAP. In Table 2, FRAP achieves significant improvements and best performance for FID and all other evaluated metrics in terms of prompt-image alignment, overall image quality, and image authenticity on the COCO-5K dataset. Especially, there is a substantial improvement in CLIP-IQA-Real score (D&B: 0.785; FRAP: 0.849) over the three other evaluated methods showing the advantage of achieving high image authenticity even when optimizing for prompt-image alignment when using our proposed FRAP method.
>
> To be able to evaluate FID, we need a dataset with reference ground truth images. We adopt the validation set of the MS-COCO dataset (Lin et al., 2014) which contains the ground-truth reference image, following the evaluation settings of works in text-to-image generation (Rombach et al., 2022; Saharia et al., 2022; Ramesh et al., 2022; Podell et al., 2023). To select prompt-image pairs that are most relevant to prompt-image alignment evaluation and specifically object presence and object-modifier binding, we only keep the relevant prompts with at least two objects and each has at least one modifier word by filtering the prompts with the spaCy (Honnibal & Montani, 2017) language dependency parser. This filtering process selects 16k most relevant prompts from the original 40k prompts in MS-COCO, and we randomly sample a 5k subset from the 16k most relevant prompts. We refer to this dataset as COCO-5K and will release this dataset to facilitate reproducibility and further research.

---

> > ### Comment · Reviewer_fVnT · 2025-02-03
> >
> > Thanks for the detailed answer to my comments. I don't have any further comments and I'm happy with the proposed changes.

---

> ### Comment · Action_Editor_vEDo · 2025-01-31
>
> Could you check out the authors rebuttal? Any further comments?

---

> ### Comment · Action_Editor_vEDo · 2025-01-31
>
> Could you check out the authors' rebuttal? Any further comments?

---

### Review · Reviewer_XQkq · 2024-12-21

**Summary Of Contributions:**

This paper introduces a novel method called FRAP for improving text-to-image (T2I) diffusion models. FRAP addresses challenges in ensuring alignment between text prompts and generated images while maintaining image realism. This approach enhances both the presence of objects and the correct binding of modifiers to objects in generated images, by leveraging a language parser to extract objects of interest and object-modifier keywords in the text prompt. FRAP then dynamically adjusts the weight of each token in the prompt during the image denoising process. FRAP also uses a combined loss function over learned cross-attention maps to optimize object presence and modifier binding.

**Audience:**

Yes

**Claims And Evidence:**

No

**Requested Changes:**

I believe the following points are critical to be addressed:

1. One of the major contributions of this work I believe lies in using prior knowledge in a language parser that brings out the object of interest and object-modifier relationship, which the proposed method then uses. However, this details wasn’t mentioned anywhere until Section 4.1. Can the authors make this stand out earlier in the paper, say, the introduction?
2. What is P in P(s, R_s) in Section 4.1? Please define. You have also used P to denote the spatial dimension of a latent map.
3. Object Presence Loss: I understand that the intuition behind this loss is to make sure that there exist some pixels in the latent map that attend to the value corresponding to an object token. Given the loss is a “max” over attention values, can the model cheat in always putting attention to objects on a single pixel? Is the Gaussian smoothing over cross-attention maps critical? Did the authors perform any ablation over this detail? Can they also provide a discussion around any implementation details that were critical?
4. Object-Modifier Binding Loss: intuitively, what the authors are proposing is that the sets of pixels attending to the object and the modifier should have high overlap. It is unclear to me how the proposed loss function in eqn. 6 would achieve that. Can the authors clarify?
5. How is adaptive prompt re-weighting relevant to the goal of generating more aligned images? It seems like an independent aspect of the method, the relevance of which is unclear to me. The authors claim in Section 4.2 that this enables their model “to generate images that are faithful to the semantics and have a realistic appearance” – what is the key insight in this strategy that would enable this?
6. Can the authors add brief descriptions of considered metrics in the appendix?
7. Quantitative comparison against baseline methods in Table 1 is seemingly not impressive. Almost all methods, proposed and baselines, seem to obtain very similar scores, on various occasions differing only on the 3rd decimal digit. For prompt-image alignment, are the utilized metrics capturing what the authors expect them to? This is very concerning to me, and without proper justification around metrics and statistically significant better results, it will be very difficult to recommend acceptance.

**Strengths And Weaknesses:**

**Strengths:**
1. The paper is well-written and easy to follow. The challenges are well-motivated and easy to grasp. The authors laid out the related works in text-to-image along with critical technical background details in Section 2.
2. Generated image samples are impressive showing more prompt-relevant images. Inclusion of both quantitative metrics and human evaluation strengthens the reliability of the results.
3. Lower inference latency and fewer UNet calls demonstrate practical advantages.


**Weaknesses:**
1. Some of the key insights, such as using prior knowledge in a language parser, are not highlighted early enough.
2. Needs more discussion and ablation around the object presence and object-modifier losses. Additionally, there is limited discussion around some implementation details, for example addressing if the Gaussian smoothing was critical.
3. Missing description of quantitative metrics.
4. Quantitative comparison against baseline methods is not impressive.

---

> ### Author Response · Authors · 2024-12-28
> **[1/2] Reply to reviewer XQkq**
>
> **[1/2] Reply to reviewer XQkq**:
>
> We thank the reviewer for reading our work and giving constructive feedback. We updated the manuscript and the changes made are in the red color (unless specified otherwise in each point below) in the manuscript. The requested changes by the reviewer are directly addressed in the comments as follows.
>
> **1. One of the major contributions of this work I believe lies in using prior knowledge in a language parser that brings out the object of interest and object-modifier relationship, which the proposed method then uses. However, this details wasn’t mentioned anywhere until Section 4.1. Can the authors make this stand out earlier in the paper, say, the introduction?**
>
> We added explanation in the introduction and Sec. 4.1 (both in red color) to make this detail stand out earlier in the paper.
>
> **2. What is P in P(s, R_s) in Section 4.1? Please define. You have also used P to denote the spatial dimension of a latent map.**
>
> We updated Sec. 4.1 (in red color) to define P(s, R_s) and related terminologies. We updated Sec. 2.1.2 to use L to represent the spatial dimension to avoid confusion.
>
> **3. Given the loss is a “max” over attention values, can the model cheat in always putting attention to objects on a single pixel? Is the Gaussian smoothing over cross-attention maps critical? Did the authors perform any ablation over this detail? Can they also provide a discussion around any implementation details that were critical?**
>
> We updated Sec.4.1.1 (in blue color) to address this point. As pointed out in the ablation studies done by Chefer et al. (2023), the Gaussian smoothing applied to the CA map is important, since it ensures that maximizing the largest activation value not only encourages a single point of high activation but rather a region of high activation values, which is critical for generating the entire object and avoiding partial generation of objects.
>
> Note that the per-token presence loss in Eq. (5) is adopted from A&E’s (Chefer et al., 2023) design. However, our design of the total object presence loss in Eq. (6) is original and different from A&E. We use the mean among all objects as the total object presence loss instead of only using the loss from the most neglected token with the lowest max activation. Our design enhances the presence of all objects instead of a single one, which is expected to be essential for complex prompts. Through our empirical evaluation in Table 6 in App. C, our design which enhances the presence of all objects outperforms A&E’s design where only the most neglected token is considered.
>
> We also consider an alternative design for the object presence loss, where we attempt to replace the object presence loss with the Total Variation (TV) loss adopted by D&B (Li et al., 2023), which encourages high activation difference across spatial locations. Through our empirical evaluation in Table 7 in App. C, the object presence loss in Eq. (5) performs better than the TV loss.
>
> **4. Object-Modifier Binding Loss: intuitively, what the authors are proposing is that the sets of pixels attending to the object and the modifier should have high overlap. It is unclear to me how the proposed loss function in eqn. 6 (Eq. (7) in the updated manuscript) would achieve that. Can the authors clarify?**
>
> We updated Sec.4.1.2 (in blue color) to address this point. Note the negative sign for L_{binding,t} in Eq. (4), which results in maximizing Eq. (7), i.e. the minimum overlap. As shown in Eq. (4), we maximize this minimum overlap between the two discrete probability distributions in Eq. (7) in a minimax fashion. Our minimax approach aims to ensure that the least probable events have high probabilities in both distributions. *This guarantees that every pixel of their respective object or modifier receives a considerable probability, regardless of which CA map is more accurate overall.* This is particularly useful in cases where there is a need to depict a small object in the scene. We found this approach to be more beneficial than Minimizing JSD (as in D&B (Li et al., 2023)) or KLD (as in SynGen (Rassin et al., 2023)), likely due to the following two reasons: (i) *it ensures a robust match between the CA maps.* This is particularly useful for representing small objects in a scene. (ii) *it is less susceptible to the influence of outliers* compared to divergence methods such as JSD and KLD. This robustness is beneficial in handling noisy and uncertain CA maps. Through our empirical evaluation in Table 8 in App. C, our object-modifier binding loss in Eq. (7) performs better than either the JSD loss or the KLD loss.

---

> > ### Author Response · Authors · 2024-12-28
> > **[2/2] Reply to reviewer XQkq**
> >
> > **[2/2] Reply to reviewer XQkq:**
> >
> > **5. How is adaptive prompt re-weighting relevant to the goal of generating more aligned images? It seems like an independent aspect of the method, the relevance of which is unclear to me. The authors claim in Section 4.2 that this enables their model “to generate images that are faithful to the semantics and have a realistic appearance” – what is the key insight in this strategy that would enable this?**
> >
> > The diffusion model's cross attention mechanism conditions on the text embedding of the text prompt to generate images that aligns with the prompt. Therefore, by altering the text embedding, we can affect the denoising process and thus affect the output latent image. Improving the prompt-image alignment means generating image that better matches the semantic of the prompt (e.g., all objects mentioned in the prompt are generated, and each have the appearances of the given modifiers).
> >
> > The manual prompt weighting technique (Hugging Face, 2023a; Stewart, 2023) described in Sec. 2.2 shows that we can manually specify per-token weight coefficients to alter the text embedding to emphasize (with a coefficient larger than 1) or de-emphasize (with a coefficient smaller than 1) the semantics behind each token to achieve the goal of improving prompt-image alignment. Take prompt "a green apple and a yellow tomato" as an example, if apple is missing or generated with incorrect color in the output image, we can increase the weight coefficient for "apple" and "green" to alter the text embedding which could lead to the generation of an image with better prompt-image alignment.
> >
> > Our method defines objective functions on the CA maps to reflect the level of object presence and level of object-modifier binding (i.e. the level of prompt-image alignment). By adaptively updating the weighting coefficients through inference-time gradient-descent guided by the objective function, we are essentially altering text embedding in the direction of strengthening object presence and encouraging object-modifier binding. The altered text embedding will then affect the output image latent though the CA mechanism to achieve the goal of generating more aligned images.
> >
> > **6. Can the authors add brief descriptions of considered metrics in the appendix?**
> >
> > We did already provide the description of metrics in Appendix A.3 of the originally submitted manuscript. During rebuttal, we also added a more brief description of these metrics in Section 5.1 of the main text (marked in red color).

---

> ### Comment · Action_Editor_vEDo · 2025-01-27
>
> Could you check out the authors' response and share any further comments?

---

> > ### Comment · Reviewer_XQkq · 2025-01-29
> > **Response to Authors**
> >
> > I thank the authors for their response. They addressed my concerns 1 through 6 regarding their presentation and justification around the chosen loss functions, and I am satisfied with their response. However, the authors did not respond to my concern (7) regarding their seemingly non-impressive results in Table 1.
> >
> > As I pointed out in my review, almost all methods, proposed and baselines, seem to obtain very similar scores differing only on the 3rd decimal digit. I further asked if the utilized metrics captured what the authors expected them to. As such, the results are very concerning to me and without statistically significant better results I cannot recommend acceptance.

---

> ### Author Response · Authors · 2025-01-29
>
> Please note that we have already replied to your concern (7) in separate official comments. Please see our comments above named "[1/2] Reply to reviewer XQkq and fVnT: regarding similar scores (or modest improvement)" and "[2/2] Reply to reviewer XQkq and fVnT: regarding similar scores (or modest improvement)".
>
> Thank you for the constructive feedback. We are glad our response addressed your concerns (1) through (6).

---

### Review · Reviewer_8tjp · 2024-12-29

**Summary Of Contributions:**

This work proposes improving existing text-to-image (T2I) diffusion models, specifically, their ability to adhere to complex prompts containing multiple entities with modifiers (adjectives) and possibly interactions (see below strengths and weakenesses) in between. The proposed method, FRAP, performs inference-time optimizations to adaptively adjust the weight of each prompt token based on pre-extracted object tokens and modifier tokens via spaCy. The optimization is carried out on the cross-attention attention map.

**Audience:**

Yes

**Broader Impact Concerns:**

This paper carries no special impact, aside from the common concerns about generative methods. However, given that this work performs no additional training to the pre-trained networks, the impact is minimal.

**Claims And Evidence:**

Yes

**Requested Changes:**

Please see the above **weaknesses** on items that may require clarification.

**Strengths And Weaknesses:**

The paper, at the time when this review was written, has already been updated per other reviewers' reviews (with text in different colours added to the main article, presumably the colours correspond to review items -- but since I cannot yet view other reviewers' comments, I am not sure). To this end, my review will be based on the already updated version.

**Strengths**
- This paper, after revision, is written with ideas conveyed clearly and concisely. I particularly appreciate the extended explanations to the method section (specifically, the loss terms and how the authors use existing tools to identify object and modifier tokens). I would imagine that without these extended explanations, the method section would be a bit more difficult to understand.
- Experimental results (qualitative ones especially) do suggest that the proposed method holds merits.

**Weaknesses**
- It would be better if the author could demonstrate the denoising process through time $t$ with and without the proposed optimization, such that one can see exactly how it influences the process. More importantly, a discussion and demonstrate of how the attention gets modified through time (and what differences would have in different timestep) would be great. Because if not mistaken, the proposed method is partially unique and novel in that, it is an online optimization that gets carried out at each timestep, rather than having a fixed weight coefficient applied on the attention map constantly (in fact the author has compared the method with such constant-weight approaches and demonstrate empirical superiority).
- Does the method account for interactions among entities? In Figure 1, the word "chasing" is labelled as a modifier for the word "dog", which is partially accurate in that it depicts the pose of the dog, but meanwhile, it should also carry an additional layer of meaning, in that the dog is "chasing" the cat, thus pointing to the "cat". However, after reading the paper my understanding would be that the proposed method treats entities individually, without accounting for the interactions or relationships among them.
- Given that the optimization is carried out at the inference time, I wonder to what extend the loss gets minimized. Is there any criteria on it, or is it determined by visual examination of whether the end results are of good quality?

---

> ### Author Response · Authors · 2025-01-12
> **[1/2] Reply to reviewer 8tjp**
>
> We thank the reviewer for reading our manuscript and providing constructive feedback. We updated the manuscript. The changes made are in the orange color in the manuscript. The items that may require clarification suggested by the reviewer are directly addressed in the comments below.
>
> **Q1. It would be better if the author could demonstrate the denoising process through time t with and without the proposed optimization, such that one can see exactly how it influences the process. More importantly, a discussion and demonstrate of how the attention gets modified through time (and what differences would have in different timestep) would be great. Because if not mistaken, the proposed method is partially unique and novel in that, it is an online optimization that gets carried out at each timestep, rather than having a fixed weight coefficient applied on the attention map constantly (in fact the author has compared the method with such constant-weight approaches and demonstrate empirical superiority).**
>
> We added Fig. 11 and discussions in App. F.1 to address this point. To demonstrate the influence of FRAP on the denoising process and see how it affects attention, we provide visualizations of the Cross-Attention (CA) maps and the intermediate images every 5 time-steps (see Fig. 11). For both Vanilla SD and FRAP, we show the CA maps for the “horse” token, the CA maps for the “turtle” token, as well as intermediate images (i.e., the denoising process) across the time steps.
>
> At t = 50 and t = 45, the CA maps for both FRAP and Vanilla SD show that the attention to the “turtle” token is less bright and sparser compared to “horse”. By using FRAP, the loss function can detect and give the feedback that “turtle” is receiving less attention (i.e., having a larger loss value). Next, FRAP adaptively adjusts the prompt weighting according to the gradients to re-emphasize the neglected object “turtle”.
>
> At t = 40 and t = 35, we can clearly observe the influence of FRAP on the denoising process and attention. The “turtle” CA maps for FRAP have a separate blob of high attention values in the bottom left corner, whereas the attention to “turtle” in Vanilla SD is smaller and overlaps with “horse”.
>
> Therefore, FRAP can adaptively adjust prompt weighting to influence the denoising process and attention, and thus improve prompt-image alignment.
>
> **Q2. Does the method account for interactions among entities? In Figure 1, the word "chasing" is labelled as a modifier for the word "dog", which is partially accurate in that it depicts the pose of the dog, but meanwhile, it should also carry an additional layer of meaning, in that the dog is "chasing" the cat, thus pointing to the "cat". However, after reading the paper my understanding would be that the proposed method treats entities individually, without accounting for the interactions or relationships among them.**
>
> Our method does account for interactions among entities through the Cross-Attention (CA) mechanism. First, this can be visualized through the CA maps in Fig. 1 for the token "chasing", where attention for "chasing" is given to both “dog” and “cat”, showing that the CA mechanism accounts for the interaction: "chasing". Second, when the modifier word is an action (e.g., chasing), the dependency parser assigns it to the word that performs the action (i.e., dog in this example). However, this does not mean that "cat" is treated individually and has no relationship/interaction to "chasing". This is because the Diffusion model still uses the original whole prompt for generation, where the information about "chasing" is passed to both "cat" and "dog" through the Cross Attention mechanism, allowing for the generation of interaction among these entities (i.e., carrying the meanings of both "dog chasing" and "cat being chased" during generation).

---

> > ### Author Response · Authors · 2025-01-12
> > **[2/2] Reply to reviewer 8tjp**
> >
> > **Q3. Given that the optimization is carried out at the inference time, I wonder to what extend the loss gets minimized. Is there any criteria on it, or is it determined by visual examination of whether the end results are of good quality?**
> >
> > We added Fig. 12 in App. F.2 to discuss this point. We visualize the trend of the loss across the time steps during inference. In Fig. 12, we show the generated images (top) and the plots of loss values across the time steps (bottom), for three different methods: Vanilla SD (left), FRAP with default settings (middle), and FRAP with a larger $\eta$ and larger $t_\text{end}$ (right).
> >
> > Referring to Vanilla SD on the left of Fig. 12, we can see that “turtle” is missing from the generated image. This is also reflected by the higher loss value of the orange curve for “turtle” compared to the lower loss value of “horse” represented by the blue curve. In other words, the curves show that Vanilla SD did not give enough attention to “turtle”, thus causing it to be neglected in the final generated image.
> >
> > In contrast, FRAP in the middle of Fig. 12 is able to bring the yellow loss curve for “turtle” down to the same level as the blue curve. During the inference-time optimization, FRAP automatically updates the prompt weighting coefficients based on the feedback gradients from the loss functions. As a result, the model gives more attention to the neglected “turtle” and can generate both objects in the final image.
> >
> > The other question is to what extent should we minimize the loss, or we should instead aim for good visual quality in the end result (i.e., the generated image). The loss roughly reflects the level of prompt-image alignment, more specifically, how well object presence and object-modifier binding are handled. By minimizing the loss, we are aiming to improve the prompt-image alignment, because the loss function provides feedback signals through the gradients so that FRAP can adaptively update the prompt weighting coefficients in the correct direction of improving prompt-image alignment during inference.
> >
> > On the right of Fig. 12, we also consider an aggressive variant of FRAP which aims to minimize the loss as much as possible, by increasing the step-size $\eta$ from 1.0 to 5.0, and changing $t_\text{end}$ to 0 (i.e., applying adaptive prompt weighting in all 51 steps, instead of only in the 25 early steps). Comparing the loss plots of the three methods, we see that this aggressive variant achieves the lowest loss values among the three methods (e.g., the orange curve reaches a very small value of around 0.2). However, the end result of the aggressive variant (right) is much worse than FRAP (middle) despite achieving lower loss values. This can be seen in the quality degradation of the grass background as well as the horse skin, which starts to look like a turtle.
> >
> > Therefore, we should not aim to minimize the loss as much as possible without considering image quality. Both image visual quality and prompt-image alignment are important for text-to-image generation, so the focus should be finding a good balance between visual quality and minimization of the loss which reflects prompt-image alignment. The step-size $\eta$ controls to what extent the loss gets minimized, as seen in Fig. 12, the loss reaches a much smaller value of around 0.2 (the orange curve in the right plot) when we increase the step-size $\eta$ from 1.0 (middle plot) to 5.0 (right plot). From our ablation experiment results on step-size $\eta$ in Table 12, we can see that increasing the step-size beyond $\eta=1$ does not increase prompt-image alignment anymore while it reduces image quality. Therefore, with the setting of $\eta=1$ used in this paper, we achieve a good balance between loss minimization (i.e., prompt-image alignment) and the visual quality of the end result.

---

> ### Comment · Action_Editor_vEDo · 2025-01-27
>
> Could you check out the authors' response and share any further comments?

---

> ### Comment · Action_Editor_vEDo · 2025-01-31
>
> Could you check out the authors' rebuttal? Any further comments?

---

> > ### Comment · Reviewer_8tjp · 2025-02-07
> > **No further comment**
> >
> > I appreciate the authors' response to the comments, I have no further questions.

---

### Review · Reviewer_1V3K · 2024-12-29

**Summary Of Contributions:**

The paper proposes a novel method called FRAP for the task of improving prompt-image alignment in text-to-image diffusion models. FRAP is an online algorithm, and does not require model fine-tuning. It works by iteratively updating prompt tokens' weights during inference timestamps based on current severity of concepts and relations between concepts and attributes (modifiers) in cross-attention maps. It shows sufficient gains in metrics over diverse datasets, not losing in general image quality and having less added inference time compared to other approaches at the same time.

**Audience:**

Yes

**Broader Impact Concerns:**

-

**Claims And Evidence:**

Yes

**Requested Changes:**

Points mentioned here would be nice to have addressed:
- Except SD-1.5, only SDXL model is considered, and there is only one table with results on SDXL, showing metrics improvements of FRAP compared to vanilla SDXL. First, it would be nice to see some qualitative evaluation examples, as it is not clear if gain in metrics is significant. Secondly, it is interesting to see if image quality degrades with FRAP on SDXL. I see this as the most significant issue of the paper, as SD-1.5 is quite old, and many new models are out, and we'd like approaches to work on them as well.
- During prompt weights optimisation, is only one optimisation step is done on each iteration? It would be nice to see ablation on number of steps and \eta value (as I see in appendix, \eta=1 in all experiments)
- It would be nice to see distributions of prompt lengths and numbers of concepts and relation pairs in datasets. Also it is interesting to see how performances of models change with increase of prompt length and numbers of concepts and relation pairs in prompt. Does the approach experience decline in quality gain? Does using spaCy somehow restrict the performance? I would actually include using spaCy to "limitations" section, as I suspect, its relation extraction is not perfect. Maybe also include some failure cases caused by spaCy.

**Strengths And Weaknesses:**

Strengths:
- Paper is clearly written, algorithm structure is well explained.
- Qualitative and quantitative results suggest that method does improve prompt-image alignment over other approaches
- Less added inference time compared to other methods in an advantage
- Ablations over loss structure are good
- Possibility to combine method with other (as Promptist) is beneficial
- It's great to see that authors included FID in updated version of the paper, as this was my main concern when reading the original version of the paper.

For the weaknesses, see "requested changes" section

---

> ### Author Response · Authors · 2025-01-12
> **[1/2] Reply to reviewer 1V3K**
>
> We thank the reviewer for reading our work and giving constructive feedback. We updated the manuscript. The changes made are in the teal color in the manuscript. The nice-to-have changes requested by the reviewer are directly addressed in the comments as follows.
>
> **Q1. Except SD-1.5, only SDXL model is considered, and there is only one table with results on SDXL, showing metrics improvements of FRAP compared to vanilla SDXL. First, it would be nice to see some qualitative evaluation examples, as it is not clear if gain in metrics is significant. Secondly, it is interesting to see if image quality degrades with FRAP on SDXL. I see this as the most significant issue of the paper, as SD-1.5 is quite old, and many new models are out, and we'd like approaches to work on them as well.**
>
> We updated Appendix D.3 to include qualitative examples on SDXL. The image quality actually improves significantly with FRAP on SDXL rather than degrades. Specifically, in Figs. 8, 9, 10, we observe significant improvements in image quality, image realness, prompt-image alignment, and the counting ability when applying FRAP to SDXL, including but not limited to the examples provided below:
>
> **Image realness and image quality.** For example, in Fig. 8, for the prompt "a black cat and a red suitcase on the street, snowy driving scene", the image generated by FRAP (2nd column, 4th row) is much more realistic and the overall image quality is also better compared to the cartoonish image generated by vanilla SDXL (2nd column, 3rd row). The same applies to prompts "a blue bird ..." and "a purple dog ..." in Fig. 8. Similarly, this can also be seen in the images for prompts "a laptop ...", "a man ...", and "a woman sitting on ..." in Fig. 9.
>
> **Prompt-image alignment.** For example, in Fig. 8, for the prompt "a green backpack and a yellow chair in the library", the image generated by FRAP (4th column, 4th row) is much better aligned with the prompt (i.e., have both "backpack" and "chair" generated in the correct color), compared to the image generated by vanilla SDXL (4th column, 3rd row) which does not have a backpack, same applies to prompts "a pink clock ..." and "a green balloon ..." in Fig. 8. Similarly, the better alignment of FRAP can also be seen in the images for prompts "a dog and a cat ...", "a little dog jumping ...", and "a woman rides her bike ..." in Fig. 9.
>
> **Counting ability.** In Fig. 10 (for the Multi-Object dataset), we frame the image in red if it has an incorrect object count. It can be seen that FRAP (12 out of 16 images are correct) generates images with a correct number of objects more often than vanilla SDXL (only 2 out of 16 images are correct). Fewer red frames are shown for FRAP.
>
> With this extensive additional qualitative evaluation on SDXL, we indeed observe gains from FRAP in realness, quality, alignment, and counting. Furthermore, the performance gain is even more significant on long and complex prompts (see answer to Q3). These extensive additional qualitative examples together with all the quantitative metrics verify that the performance benefit of FRAP is convincing, robust, not random, and is not only limited to SD1.5 model.
>
> **Q2. During prompt weights optimisation, is only one optimisation step is done on each iteration? It would be nice to see ablation on number of steps and \eta value (as I see in appendix, \eta=1 in all experiments)**
>
> In this response to this, we have added Table 11 and Table 12 in the Appendix to provide additional ablation experiment results on the number of optimization steps and the $\eta$ value (i.e., step-size) respectively.
>
> In Table 11, we present an ablation study on the number of optimization steps performed at each time-step, comparing a range of values from 1 to 6. From the results in Table 11, we confirm that the setting used in this paper, i.e., 1 optimization step, achieves the best performance in image authenticity (realness) and overall image quality, while achieving lower latency and comparable prompt-image alignment capability.
>
> In Table 12, we present an ablation study on the choice of step-size $\eta$. By comparing results on a range of $\eta$ values from 0.2 to 5.0, we observe that increasing $\eta$ improves the prompt-image alignment, whereas decreasing $\eta$ favors the image quality and image authenticity. Results show that the setting of $\eta = 1$ used in this paper achieves a good balance between prompt-image alignment, overall image quality, and image authenticity.

---

> > ### Author Response · Authors · 2025-01-12
> > **[2/2] Reply to reviewer 1V3K**
> >
> > **Q3. It would be nice to see distributions of prompt lengths and numbers of concepts and relation pairs in datasets. Also it is interesting to see how performances of models change with increase of prompt length and numbers of concepts and relation pairs in prompt. Does the approach experience decline in quality gain? Does using spaCy somehow restrict the performance? I would actually include using spaCy to "limitations" section, as I suspect, its relation extraction is not perfect. Maybe also include some failure cases caused by spaCy.**
> >
> > In response to the reviewer's suggestion, we analyzed the statistics of the datasets and updated Table 4 in Appendix to include the average prompt length (i.e., average number of tokens), average number of objects tokens, and average number of relation pairs per each prompt for each dataset. As seen in Table 4, our evaluation is performed on a diverse set of datasets with varying complexities (i.e., ranging from 5 to 12.1 tokens per prompt, from 2 to 4.2 object tokens per prompt, and from 0 to 3.1 relation pairs per prompt).
> >
> > We sort Table 5 according to the average prompt length (i.e., dataset with shorter prompts is listed at the top, and dataset with longer and more complex prompts listed at the bottom). A general trend is that longer prompts also tend to include more subject tokens and more relation pairs. We can see in Table 5 that as prompt length and complexity increases, our method actually demonstrates an even larger quality gain rather than a decline. For example, FRAP achieves much higher realness on the CLIP-IQA-Real metric (realness) on the two most complex datasets Color-Obj-Scene (FRAP: 0.835, D&B: 0.808) and COCO-5K (FRAP: 0.849, D&B: 0.785). This shows the importance of using FRAP especially for long and complex prompts (with more tokens, concepts, and relation pairs), where existing methods suffer from significant degradations in image realness.
> >
> > In response to the reviewer's suggestion, we have added "using spaCy" to the limitations section (see App. E). To automatically extract objects and modifiers from the prompt, we choose the spaCy parser since it is a lightweight and robust framework dedicated to many common NLP tasks such as dependency parsing, entity recognition, and sentence segmentation which is not a bottleneck technology nowadays in NLP and can also be done by other methods like Stanza and NLTK. In this work, we have shown end-to-end performance benefits of FRAP (as a whole stack of inference-time optimization) in an extensive range of settings, while the specific choice of dependency parser is orthogonal to our main technical contribution and focus here.
> >
> > That being said, spaCy and any other parsers are not perfect and may occasionally omit a modifier, for which we have provided some examples. However, this is not fatal to the generation process of the Diffusion Model (DM), which still relies on the original whole prompt for generation. Specifically, spaCy-identified modifiers are only used to index the cross-attention maps when calculating the binding loss function in the inference-time prompt reweighting optimization. A word in the prompt, even if omitted by spaCy, is still incorporated in the generation process by the DM. For example, in the prompt “A dog and a cat curled up together on a couch”, spaCy did not extract “curled” as a modifier. However, FRAP can still robustly generate images with the correct semantics of “curled” (see 2nd row, 2nd column in Fig. 9) since the information of “curled” can be passed into the “cat” and “dog” in the image through the cross-attention mechanism of the original Diffusion Model. Therefore, FRAP does not critically depend on the exact accuracy of modifier extraction and is robust to failure cases of spaCy.

---

> ### Comment · Action_Editor_vEDo · 2025-01-27
>
> Could you check out the authors' response and share any further comments?

---

> > ### Comment · Reviewer_1V3K · 2025-02-07
> >
> > Sorry for the late reply, I am satisfied with the author's responses, I have no further questions

---

### Author Response · Authors · 2024-12-28
**[1/2] Reply to reviewer XQkq and fVnT: regarding similar scores (or modest improvement)**

**[1/2] Reply to reviewer XQkq and fVnT: regarding similar scores (or modest improvement):**

>"Quantitative comparison against baseline methods in Table 1 is seemingly not impressive. Almost all methods, proposed and baselines, seem to obtain very similar scores, on various occasions differing only on the 3rd decimal digit. For prompt-image alignment, are the utilized metrics capturing what the authors expect them to? This is very concerning to me, and without proper justification around metrics and statistically significant better results, it will be very difficult to recommend acceptance."

> "the quantitative results, while promising, demonstrate only a modest improvement over baseline methods. The authors should address this directly, discussing the significance of these gains"

In fact, we disagree with this claim that our results demonstrate only a slight improvement over baseline methods (or all methods achieve similar scores according to Table 1). We want to explain how the numerical results in Table 1 should be read and interpreted as well as their significance by reviewing all metrics and datasets used here again. We are confident that we have provided extensive evaluation by thoroughly evaluating a range of realness, quality and alignment metrics (not only looking at one or two IQA metric in the table) on several harder or easier datasets. These results are not demonstrating only modest improvement over baseline methods. Neither is “prompt-alignment metric” the sole or main advantage of FRAP over existing baselines.

First, we explain the metrics and their meanings. To assess generation quality thoroughly, we consider 3 aspects in evaluation and comparison:

1. *Generation authenticity and realness (i.e. CLIP-IQA-Real), which is our main problem to solve as stated in the introduction (which existing methods fail to achieve).*
    - We adopt the "real" component of the CLIP-IQA metric (i.e. CLIP-IQA-Real) which specifically reflects the level of image authenticity, i.e., how realistic the generated images look.
2. *Prompt-image alignment (i.e .TTS, Full-CLIP, and MOS), which is also our general goal to achieve just like existing methods aim to (but sacrificing realness).* Each metric (i.e. TTS, Full-CLIP, MOS) reflects how well the generated image aligns with the given prompt at a different semantic level:
    - TTS evaluates the sentence-to-sentence level alignment (between the original text prompt and the generated caption of the synthesized image).
    - Full-CLIP reflects the sentence-to-image level alignment (between the original text prompt and synthesized image).
    - MOS considers a finer-grained alignment between sub-sentence (i.e., part of the prompt) and the synthesized image.
3. *General and overall image quality (e.g., image sharpness, brightness), which is the metric we aim to maintain with FRAP (and thus showing similar performance to prior works). To assess general image quality, we utilize:*
    - HPSv2, which is trained on human preference datasets to reflect human preference for overall image quality
    - CLIP-IQA, which is the overall CLIP-IQA score averaged over 4 criteria including “quality”, “noisiness”, “natural”, and “real”.

The detailed description of all these evaluation metrics can be found in Appendix A.3.

Existing methods aim to improve prompt-image alignment, but they sacrifice the image authenticity (how real the generated image looks) which can be seen from their low “CLIP-IQA-Real score” in Table 1, which is the most important score to look at. For example, they often generate cartoonish and unrealistic images (see Fig.3, where images generated by A&E and D&B have cartoonish appearances in "turtle", "horse", and "clock"), since their specific latent code optimization techniques could drive the latent code go out-of-distribution (OOD). Less real appearance and looking suffered by existing methods (demonstrated by a low “CLIP-IQA-Real” score) is the main issue FRAP aims and manages to avoid.

---

> ### Author Response · Authors · 2024-12-28
> **[2/2] Reply to reviewer XQkq and fVnT: regarding similar scores (or modest improvement)**
>
> **[2/2] Reply to reviewer XQkq and fVnT: regarding similar scores (or modest improvement):**
>
> The takeaway from FRAP’s evaluation (Table 1) is that it significantly improves image realness and authenticity, in other words, generating true-to-life images with realistic appearance, while enhancing prompt-image alignment and maintaining general image quality at the same time. This is demonstrated by Table 1, where FRAP achieves significant improvement in real appearance and image authenticity (demonstrated by much higher CLIP-IQA-Real scores). Second, FRAP also enhances prompt-image alignment on challenging datasets (i.e., Color-Obj-Scene and COCO-Attribute) that have modifier wording to the objects. FRAP shows slight prompt-image alignment improvement on the easier COCO-Subject dataset which does not involve any modifier wording, since it is easier to handle with all methods. Finally, FRAP still maintains overall image quality by achieving either comparable or slight improvements in HPSv2 and CLIP-IQA, which assess general noiseness and quality (not specifically realness in looking), while enjoying a lower latency than A&E and D&B. We also updated Table 1 to include statistical significance t-test results. Therefore, we cannot just look at a few specific numbers we are trying to only maintain. The gist is that FRAP always enhances realness significantly and often enhances alignment, while maintaining other overall image quality, which means it is a very robust method.
>
> Furthermore, assessment based on visual quality comparisons shows that by achieving a higher CLIP-IQA-Real, FRAP can generate much more real-looking images and makes a big difference in visual quality than existing methods. In fact, images with similar overall/general image quality can have a dramatic visual difference due to different CLIP-IQA-Real scores. For example, if we compare the images in Figure 3 generated by A&E (5th column, 2nd row) and by FRAP (5th column, 4th row) under the text prompt “a horse and a bird on the bridge”, we find that although the overall image quality is close for both methods (with HPSv2 scores of 0.271 and 0.278), there is a substantial gap in the CLIP-IQA-Real score (0.672 for A&E and 0.987 for FRAP). Therefore, FRAP’s generated image looks much better visually than A&E (i.e., comparing the real-looking "horse" generated by FRAP against the unrealistic "horse" generated by A&E).
>
> In fact, a further look at all the examples in Figure 3 and Figure 4 on all prompts reveals that D&B and A&E are making generated images artificial in looking, which is even worse than original SD, although their reported overall quality (HPSv2 and CLIPIQA) is fine by numerical results in Table 1. However, the gist is that although A&E and D&B aim to solve the alignment issues and achieves reasonable numbers, their methods inherently suffers OOD issue and makes the generation unreal, which is basically the problem we are trying to fix, as stated in the introduction. As a result, the proposed method FRAP achieves much higher realness score (CLIP-IQA-Real) than these existing methods, while also improving alignment on two more challenging datasets and maintaining all the other general image quality. Visually, it generates much better and real-looking images as shown in Figure 3,4. On the easier COCO-Subject dataset, FRAP is also not losing. With all these thorough and extensive results (based on a range of metrics and datasets) viewed together, we are showing FRAP is a robust method and has a significant value in promoting both prompt alignment and general realness simultaneously, which current methods fail to achieve.

---

### Decision · Action_Editor_vEDo · 2025-02-22

**Recommendation:** Accept as is

**Comment:**

The paper proposed an adaptive prompt weighting method to improve text-to-image generation models' alignment between prompts and generated images. Reviewers liked the work in general, but two reviewers had reservations about the technical novelty and experimental results. Hence, the decision is to recommend the paper for acceptance and yet no presentation in ICLR.

**Audience:**

Yes. Researchers working on diffusion models, image generation, and alignment will find the work interesting.

**Claims And Evidence:**

Yes, the reviewers had some concerns and questions about the clarity of the approach, intuition, and results, and the authors addressed them well in the revision (except the results, about which the last reviewer still had reservation post-rebuttal).